# Jr. AI Scientist and Its Risk Report:
# Autonomous Scientific Exploration from a Baseline Paper

**Atsuyuki Miyai**\*                                              *miyai@cvm.t.u-tokyo.ac.jp*
*The University of Tokyo*

**Mashiro Toyooka**\*                                             *toyooka@hal.t.u-tokyo.ac.jp*
*The University of Tokyo*

**Takashi Otonari**                                              *otonari@cvm.t.u-tokyo.ac.jp*
*The University of Tokyo*

**Zaiying Zhao**                                                 *zhao@cvm.t.u-tokyo.ac.jp*
*The University of Tokyo*

**Kiyoharu Aizawa**                                              *aizawa@hal.t.u-tokyo.ac.jp*
*The University of Tokyo*
*Tokyo University of Science*

**Reviewed on OpenReview:** *https://openreview.net/forum?id=OeVO62d8Sw*

## Abstract

AI Scientist systems are autonomous agents capable of conducting scientific research. Understanding their current capabilities and risks is essential for ensuring trustworthy and sustainable AI-driven scientific progress while preserving the integrity of the academic ecosystem. To this end, we develop Jr. AI Scientist, a state-of-the-art autonomous AI scientist system that mimics the core research workflow of a novice student researcher: Given the baseline paper from the human mentor, it analyzes its limitations, formulates novel hypotheses for improvement, validates them through rigorous experimentation, and writes a paper with the results. Unlike previous approaches that assume full automation or operate on small-scale code, Jr. AI Scientist follows a well-defined research workflow and leverages modern coding agents to handle complex, multi-file implementations, leading to scientifically valuable contributions. Through our experiments, the Jr. AI Scientist successfully generated new research papers that build upon real NeurIPS, IJCV, and ICLR works by proposing and implementing novel algorithms. For evaluation, we conducted automated assessments using AI Reviewers, author-led evaluations, and submissions to Agents4Science, a venue dedicated to AI-driven scientific contributions. The findings demonstrate that Jr. AI Scientist generates papers receiving higher review scores by DeepReviewer than existing fully automated systems. Nevertheless, we identify important limitations from both the author evaluation and the Agents4Science reviews, indicating the potential risks of directly applying current AI Scientist systems and key challenges for future research. Finally, we comprehensively report various risks identified during development. We believe this study clarifies the current role and limitations of AI Scientist systems, offering insights into the areas that still require human expertise and the risks that may emerge as these systems evolve. The generated papers and the authors' annotations on them are included in the supplementary materials. Issues, comments, and questions are all welcome in `https://github.com/Agent4Science-UTokyo/Jr.AI-Scientist`.

---

\*Equal contribution.

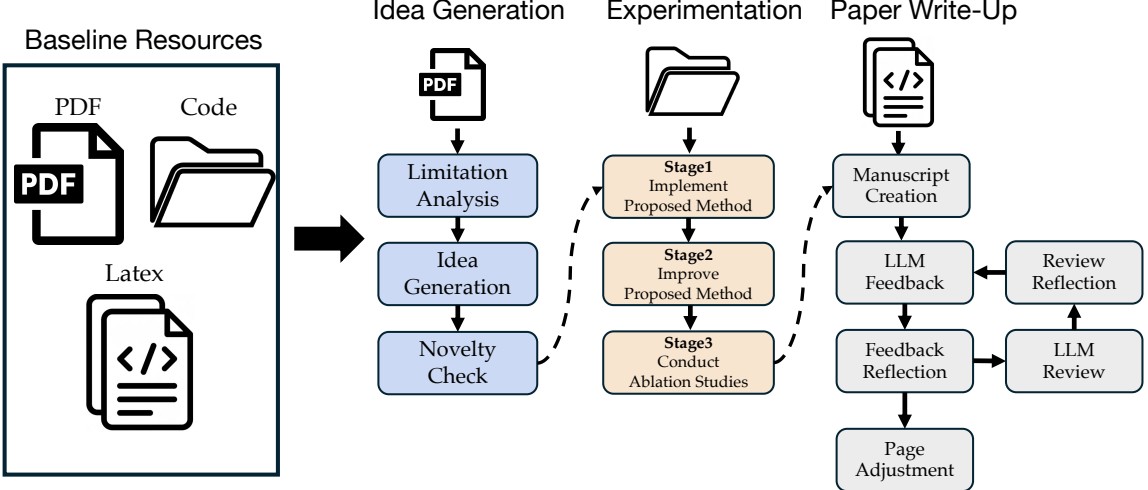

Figure 1: **Jr. AI Scientist Workflow.** We provide the baseline paper, its LaTeX source files, and the associated codebase. By effectively utilizing these resources across all phases, the system significantly improves the quality of the generated paper.

## 1 Introduction

Understanding the current upper bound of capabilities in AI Scientist systems, autonomous agents capable of conducting scientific research, is crucial for promoting sustainable, AI-driven scientific progress. Nevertheless, developers must remain conscious of the potential risks these systems pose to the academic ecosystem and commit to advancing them responsibly. Since 2025, a new venue dedicated to evaluating AI-driven scientific contributions, the Agents4Science conference (Zou et al., 2025), has emerged. Through such a platform, developers of AI Scientist systems are encouraged to engage in responsible research and development, ensuring both the protection of the academic ecosystem and the long-term sustainability of scientific progress.

In recent years, several works have explored the concept of AI Scientists (Lu et al., 2024; Weng et al., 2025a; Yamada et al., 2025; Tang et al., 2025). However, the quality of research papers produced by these systems remains insufficient. One major reason is that the problem setting of achieving fully automated science is overly ambitious and often lacks clearly defined scientific goals for AI Scientists. Without a specific goal, these systems tend to generate undirected discoveries that appear to lack genuine scientific value. Another limitation is that current systems are limited to small-scale code experiments (Zhu et al., 2025b; Lu et al., 2024; Yamada et al., 2025), lacking the scale and complexity needed for meaningful science. Achieving real scientific contributions requires not just ideas but strong implementation capability to handle complex codebases.

As an initial step toward enabling AI Scientists to produce genuine scientific value, we can take inspiration from how student researchers begin their research. When a student first joins a research lab, a common and meaningful process is as follows: the mentor assigns a key paper, the student analyzes its limitations, proposes an improvement hypothesis, implements the idea on the baseline code, validates the hypothesis through thorough experiments, and finally writes a paper summarizing the results. Through this process, the student learns the fundamental workflow of scientific research and gains the skills and experience needed for more creative work later on. Also, improving a baseline method is not only an important stage in early research training but also a valuable research goal in many fields where advancing task performance remains a central scientific pursuit.

In this paper, we introduce Jr. AI Scientist, a new AI Scientist that mimics the essential research workflow of a novice student researcher: Starting from a baseline paper, it identifies key limitations, proposes an improvement hypothesis, validates it through rigorous experimentation, and writes a paper with the results. Specifically, A Jr. AI Scientist is defined as an AI Scientist that is given baseline resources and focuses on extending the baseline. This setting is novice student analog, where a student builds upon an existing paper

Table 1: Comparison of the starting point, code complexity, and review scores among existing AI Scientist systems. Previous methods often made overly ambitious assumptions in their problem formulation and were limited to handling only simple, single-file codebases, which resulted in significantly lower review scores. In contrast, our system can substantially improve review scores by utilizing the baseline paper and its associated codebase.

| AI Scientist Systems | Starting Point | Code Complexity | Review Score |
|---|---|---|---|
| AI Scientist-v1 (Lu et al., 2024) | Template code | Single file | 3.30 |
| AI Scientist-v2 (Yamada et al., 2025) | General idea | Single file | 2.75 |
| AI Researcher (Tang et al., 2025) | 15-20 existing papers | Multiple files | 3.25 |
| Jr. AI Scientist (ours) | One baseline paper and code | Multiple files | **5.75** |

which the mentor gave. The workflow of Jr. AI Scientist is shown in Figure 1. We provide the baseline paper, its LaTeX source files, and the associated codebase for each stage. Table 1 shows a comparison of the problem setting with existing research. This problem setting reframes previously ambitious goals into a more specific objective, providing a clear optimization direction for the AI Scientist. Moreover, because the framework operates on the actual codebase of baseline papers, it can generate results with genuine scientific value. These aspects collectively represent an essential first step toward the autonomous generation of reliable and high-quality research papers.

Our Jr. AI Scientist consists of three main components: (1) automatic idea generation based on the limitations of a given paper, (2) automatic implementation and thorough validation of the proposed ideas, and (3) automatic writing of a research paper based on the obtained results. This system is built upon AI Scientist v2 (Yamada et al., 2025), but our work differs from prior studies (Lu et al., 2024; Weng et al., 2025a; Yamada et al., 2025; Tang et al., 2025) in several key aspects: First, by leveraging the latest coding agents (e.g. Claude Code (Anthropic, 2025)), our system can perform meaningful improvements and edits on real multi-file codebases, which were challenging for previous AI Scientist systems. Second, by incorporating the full set of resources from a given baseline paper, the system exploits all available artifacts such as LaTeX sources, PDFs, and codebases, thereby substantially improving the scope and quality of every stage in the research pipeline. Finally, by refining every stage of the process, our framework enables the autonomous generation of research papers that are both higher in quality and more trustworthy.

For the baseline papers, we selected papers for which we obtained permission from the original authors. Specifically, we used three papers: NeurIPS 2023 paper (Miyai et al., 2023) and IJCV 2025 paper (Miyai et al., 2025b) on out-of-distribution (OOD) detection (a task that aims to detect semantic classes outside the predefined set of semantic classes) (Hendrycks & Gimpel, 2017; Yang et al., 2024), and an ICLR 2025 spotlight paper (Zhang et al., 2025) on pre-training data detection for large language models (LLMs). Refer to Section 4.1 for the detailed rationale behind the selection of these papers. For the evaluation, we conducted three evaluations: (1) an automated assessment using DeepReviewer (Zhu et al., 2025a), (2) an author-led evaluation, and (3) submission to the Agents4Science conference (Zou et al., 2025). DeepReviewer automatically compared our generated papers with existing AI-generated works to assess overall quality. The author evaluation examined the outputs for hallucinations or fabricated content. Finally, the Agents4Science (Zou et al., 2025) platform provides rigorous evaluation and feedback from the community platform.

Through our experiments, the Jr. AI Scientist successfully generated new research papers that build upon the above top-tier venues works by proposing and implementing novel algorithms. As for the evaluation using DeepReviewer (Zhu et al., 2025a), the papers generated by Jr. AI Scientist achieved substantially higher review scores compared to the existing AI-generated papers. Therefore, our Jr. AI Scientist can be regarded as the most capable autonomous AI Scientist. However, we also observed that Jr. AI Scientist still exhibits some failures and unresolved challenges through the author evaluation and Agents4Science conference. To share these challenges and lessons with the research community, we analyze the feedback and evaluation results from the Agents4Science conference and include the author evaluation, which helps clarify what is required to further improve the quality of Jr. AI Scientist systems.

Finally, we perform an in-depth report of the risks encountered during the development of our system. Although few existing studies have provided a comprehensive discussion of these issues, we believe that accurately documenting such risks is essential to avoid overestimating current AI Scientists' capabilities and to build a clear understanding of their remaining challenges. Our risk report highlights several critical issues, including the potential for review-score hacking and difficulties in ensuring proper citation, interpreting results, and detecting fabricated descriptions. We believe that these findings provide valuable guidance on the potential risks that exist both in the current AI Scientist research and as this field continues to grow. Through this comprehensive report, we aim to foster a deeper understanding of current AI Scientist systems and contribute to their safe and trustworthy development.

Our contributions are summarized as follows:

- **Development of a New AI Scientist**: We developed Jr. AI Scientist, a new system that starts from a baseline paper and its associated codebase, and is capable of handling complex, multi-file implementations, overcoming a major limitation of previous AI Scientist systems.

- **Revealing Strengths and Limitations of Jr. AI Scientist**: We conducted extensive evaluations using open-source AI reviewers, Agents4Science, and author evaluation. The results demonstrate that Jr. AI Scientist generates higher-quality research papers than existing AI Scientists, while also revealing key challenges for future improvement.

- **Thorough Risk Report**: We report the observed risks throughout the project. We believe these reports offer insights into the areas that still require human expertise and the risks that may emerge as these systems evolve.

## 2 Related Work

**Automated Scientific Discovery.** Recent progress has significantly reshaped the role of AI in automating end-to-end scientific research. AI Scientist-v1 (Lu et al., 2024) was an early milestone, showcasing how advanced language models can autonomously generate research ideas, run experiments, and draft scientific papers. This work was followed by a series of subsequent studies in machine learning fields (Intology, 2025; Tang et al., 2025) and diverse scientific disciplines (Villaescusa-Navarro et al., 2025; Mitchener et al., 2025) that further advanced this line of research. However, these approaches often suffer from an overly ambitious problem setting that aims to achieve fully automated science and tend to lack clearly defined scientific objectives for AI Scientists. Without specific goals, such systems often produce undirected discoveries that lack genuine scientific value. To address this issue, our Jr. AI Scientist builds on existing baselines and conducts research within a well-defined research workflow, aiming to generate higher-quality scientific papers. As concurrent work, DeepScientist (Weng et al., 2025b) also adopts a baseline-based approach. DeepScientist (Weng et al., 2025b) focuses mainly on experimental performance, formalizing discovery as a Bayesian Optimization problem. However, its overall framework design and workflow integration are outlined at a high level, with limited discussion of implementation details. In contrast, our Jr. AI Scientist explicitly articulates each stage of the research process and further aims to contribute to the community by comprehensively reporting the failures and risks encountered throughout scientific exploration.

**AI-Assisted Scientific Research.** Research specialized for each element of the research process has also been actively explored (Chen et al., 2025). For the idea generation phase, Si et al. (2025b) investigates the novelty of the LLM-generated ideas, and Si et al. (2025a) investigates the ideation–execution gap. For the survey phase, OpenScholar (Asai et al., 2026) have been developed to support literature review. For the experimental phase, AlphaEvolve (Novikov et al., 2025) leverages large-scale trial-and-error strategies to enhance the performance. For the writing and review phase, CycleResearcher (Weng et al., 2025a) provides a learning framework specialized for scientific writing, while DeepReviewer (Zhu et al., 2025a) focuses on the review process. Recent studies provide a comprehensive overview of automated review, outlining key challenges, proposing a practical review pipeline for real-world implementation, and constructing a large-scale dataset to support automated review research (Zhuang et al., 2025; Lin et al., 2023a;b). Beyond these, rather than pursuing full automation like AI Scientists, AI Co-Scientist (Gottweis et al., 2025) emphasizes

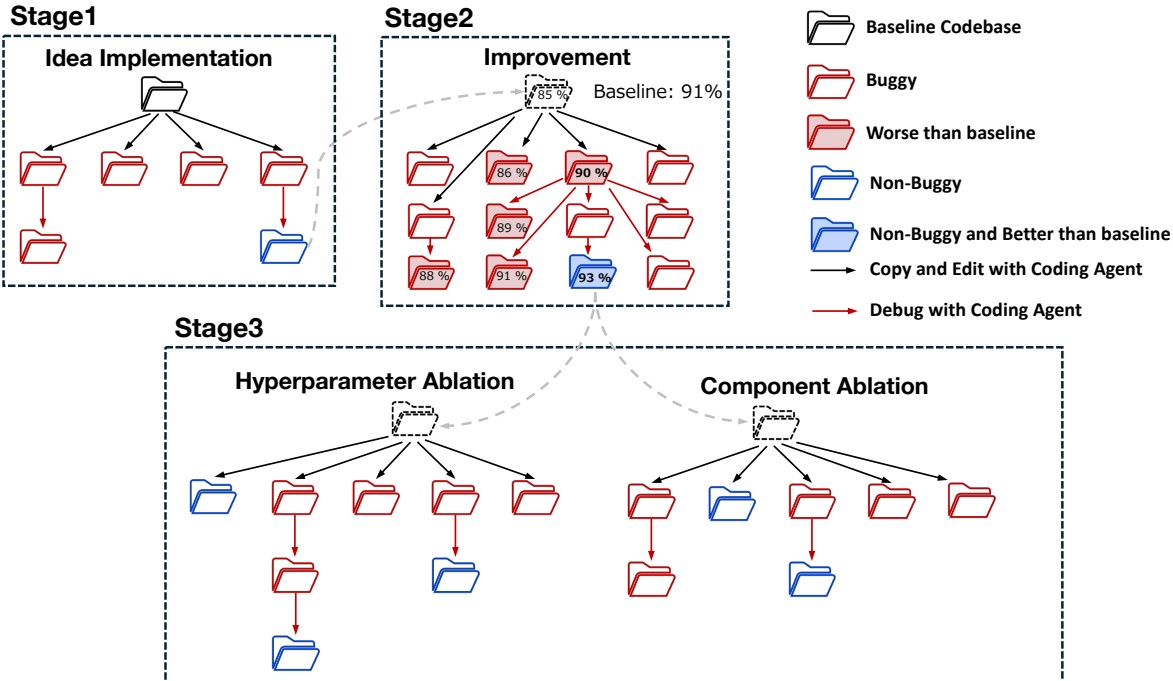

Figure 2: **Jr. AI Scientist Workflow for the Experiment Phase.** The workflow consists of three stages. Through bug management and performance tracking, our system passes the most promising experimental nodes to the next stage.

collaboration between humans and AI. In this work, instead of focusing on individual parts of the research process, we investigate the entire end-to-end research cycle, aiming to rigorously evaluate both the performance and the limitations.

**Failures and Risk Analysis for AI Scientist Systems.** There are very few studies that thoroughly analyze or report the risks and failure cases of AI Scientist systems. Although Tang et al. (2024) summarizes the risks that AI Scientists may pose, it focuses on hypothesis-based potential risks, rather than empirically observed risks or failures. While Beel et al. (2025) provides an in-depth analysis of failure cases in AI Scientist-v1 (Lu et al., 2024), these findings are based on the early AI Scientist, and the analysis was not conducted from the developer's perspective. While Luo et al. (2025) examines four failure modes (benchmark selection, data leakage, metric misuse, and post-hoc bias), their analysis is limited to experimental diagnostics and early AI Scientists without modern coding. Thus, their analysis remains somewhat limited in scope.

Therefore, we will provide a more comprehensive report on the various risks identified during the development of our state-of-the-art AI Scientist, in order to deepen the community's understanding of AI Scientists.

# 3 Jr. AI Scientist

In this section, we describe the mechanisms behind the three components of the Jr. AI Scientist: idea generation, experimentation, and writing. First, in Section 3.1, we describe the necessary preparation. Next, in Section 3.2, we explain the methods for idea generation. Next, in Section 3.3, we discuss how agents can execute and manage experiments. Finally, in Section 3.4, we explain the writing process.

## 3.1 Preparation: Baseline Paper Selection

The preparation stage involves selecting a baseline paper, obtaining its LaTeX source files and PDF, and the baseline code. This setup is realistic because many recent publications are released on arXiv with LaTeX sources, and their implementation code is shared on GitHub. While there might be some augment that AI agents should automatically select a baseline and reproduce the code, current reproducibility rates from papers

are still limited (Siegel et al., 2025; Xiang et al., 2025; Starace et al., 2025), making complete automation premature. Since our goal is to emulate how a human scientist engages in early-stage research under the guidance of a mentor, we explicitly include this preparatory stage.

When constructing the baseline code, we followed AI Scientist v1 (Lu et al., 2024) and made only minor modifications to the existing implementation so that the experiments could be executed via `baseline.py` and the results could be visualized via `plot.py`. Defining such an experimental entry point facilitated easier management and reproducibility of the execution process in the experimental section.

### 3.2 Idea Generation Phase

We provide a baseline paper as text to an LLM (e.g. o4-mini (OpenAI, 2025b)) and prompt it to output the limitations of the work. Based on both the baseline paper and the limitations, the LLM is then guided to propose potential research ideas. Following AI Scientist v2 (Yamada et al., 2025), the system evaluates the originality of proposed ideas through literature review tools such as Semantic Scholar, which review papers citing the baseline work and papers with similar concepts. If conceptually similar ideas are identified, they are refined; otherwise, they are clearly distinguished from prior work. These steps define the preliminary research idea.

### 3.3 Experiment Phase

The Experiment Phase mainly involves implementing and iterating on the implementations through experiments. It is divided into three stages: Stage 1: Idea Implementation, Stage 2: Iterative Improvement, and Stage 3: Ablation Study. The workflow of each stage is shown in Figure 2. We first describe the general procedure for using the coding agent, followed by an explanation of the implementation at each stage.

#### 3.3.1 Preliminary: Coding Agent Usage

A powerful coding agent (e.g. Claude Code (Anthropic, 2025)) is employed to translate research ideas into concrete implementations. We provide the coding agent with a working directory that contains the baseline implementations (prepared at Section 3.1) and give it detailed instructions through input prompts. The agent is informed of how to use the main scripts—`baseline.py`, which serves as the experimental entry point, and `plot.py`, which visualizes the experimental results. We use `claude-sonnet-4-20250514` within Claude Code (version 1.0.24).

The coding agent is allowed to read and write any files within the working directory. For efficient directory exploration, it is permitted to use commands such as `ls` and `grep`, while commands that may cause side effects (e.g. `python` or other execution commands) are not allowed. After the agent generates a runnable file (e.g. `proposed_method.py` in Stage1), our system mechanically executes the specified command. The coding agent is generally given up to 30 turns to complete each assigned task.

#### 3.3.2 Stage1: Idea Implemention

The system manages four experimental nodes running in parallel, each responsible for implementing and testing a proposed idea independently. Within each node, the coding agent receives the baseline code and a research idea, and writes a directly executable script named `proposed_method.py`. Once the coding agent finishes writing the implementation, the system sequentially runs `proposed_method.py` and `plot.py`. If a result file is successfully generated, the codebase is marked as Non-Buggy; otherwise, it is labeled as Buggy If a visualization image is also produced, it is further marked as Non-Plot-Buggy; otherwise, as Plot-Buggy. Each iteration of this process, coding and execution, is counted as one trial and is repeated until a bug-free implementation is obtained.

As shown in Figure 2, if any node completes successfully without encountering bugs, its codebase is carried forward to Stage 2. During execution, four experimental nodes are selected. If all currently running nodes fail, the system selects the next nodes, either by initializing new nodes from the baseline code or by debugging previously generated buggy codebases. When debugging buggy codebases, we provide the coding agent with

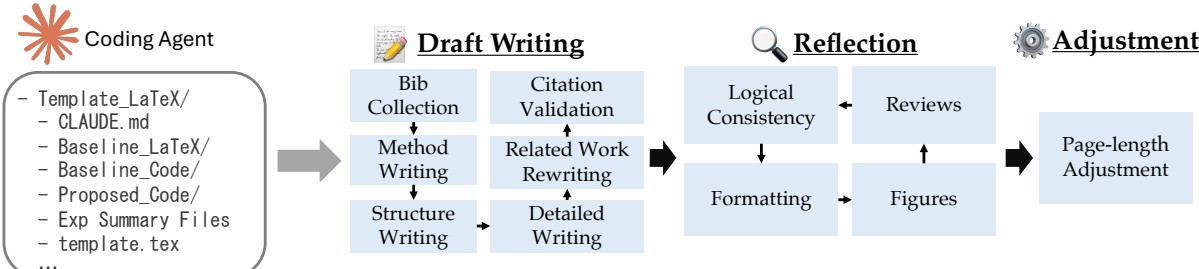

Figure 3: **Jr. AI Scientist Workflow for the Writing Phase.** The Writing process consists of three steps: Draft Writing, Reflection, and Adjustment.

detailed runtime feedback, such as standard output and error messages, to guide iterative debugging until the issue is resolved. We set this stage to run for 12 iterations.

### 3.3.3 Stage2: Iterative Improvement

Stage 2 focuses on iteratively improving the method implemented in Stage 1 until its performance metrics surpass those of the baseline. In each trial, the coding agent first proposes an improvement idea to the experimental code, and then applies the modification based on the improved idea. To avoid overwriting the Stage 1 results, we instruct the coding agent to use a new entry file named `improved_proposed_method.py` as the implementation target. The system then executes this script, followed by `plot.py`, to generate results and visualizations in the same manner as Stage 1.

As shown in Figure 2, for each trial, the experimental codebase is selected probabilistically from either (1) the Stage 1 implementation or (2) the node containing the best-performing implementation observed so far. Stage 2 ends when a bug-free implementation surpasses the baseline performance. The resulting code is then passed to Stage 3. We set this stage to run for 50 iterations.

### 3.3.4 Stage3: Ablation Study

Stage 3 performs ablation studies on the improved method implemented in Stage 2. In each trial, the system uses an LLM to generate ablation study ideas and then employs the coding agent to implement corresponding scripts based on those ideas. To encourage higher-quality ablation ideas, we first instruct the coding agent to produce a textual description of the Stage 2 method, which is then provided to the LLM as context for generating more meaningful ablation ideas. The generated ablation ideas include hyperparameter ablations, which analyze the sensitivity of the method to key hyperparameters, and component-level ablations, which assess the contribution of each component to the overall performance. To avoid overwriting Stage 2's code, we instruct the coding agent to use new entry files named `hyperparam_ablation_study.py` and `component_ablation_study.py` as the implementation target. The iterations are executed until a sufficient number of experimental results are obtained.

### 3.4 Writing Phase

We primarily used a coding agent (e.g. Claude Code (Anthropic, 2025)) as a writing agent for the writing process. As shown in Figure 3, the Writing Phase consists of three stages—Draft Writing, Reflection, and Adjustment. Below, we describe the resources provided to the writing agent and the details of each stage.

### 3.4.1 Preliminary: Resources Provided to Writing Agent

**Conference LaTeX Template.** We provide the conference LaTeX template to the writing agent. We give the Agents4Science template, and the corresponding directory is set as the working directory where the writing agent operates.

**Instruction Markdown File for the Writing Agent.** An instruction file in Markdown format is provided to the writing agent. This document defines the overall structure of the paper, outlining how each section

should be organized, and offers detailed guidelines on the key points and considerations for writing each part of the manuscript.

**Baseline LaTeX Files and Code.** We provide the writing agent with the baseline LaTeX files and code. These resources are mainly used to explain the baseline method in the Method section.

**Stage 2 Proposed Method Code.** The writing agent is also given the Stage 2 code (the proposed method). This is primarily referenced in the Method section when explaining the proposed approach.

**Experiment Summaries for Each Stage.** Following the protocol of AI Scientist v2, we provide the writing agent with summarized JSON files containing key experimental results for each stage (`baseline_summary.json`, `improved_research_summary.json`, `component_ablation_summary.json`, and `hyperparam_ablation_summary.json`). These files include essential information such as experimental descriptions, results, and paths for visualization results, which are crucial for the writing agent when writing the experimental results section. In addition, for ablation studies, we not only provide the JSON files but also automatically convert them into LaTeX table files (`component_ablation_summary_table.tex` and `hyperparam_ablation_summary_table.tex`). This conversion has significantly reduced numerical transcription errors in the paper.

The only materials that humans need to prepare for each paper generation are the LaTeX source files and the baseline codebase of the baseline paper. The CLAUDE.md file and the conference LaTeX template are shared across all experiments. The code and experimental files for the proposed methods are automatically generated during the experimental stage. Using these inputs, the AI agent automatically writes all sections and captions in the paper.

### 3.4.2 Draft Writing

As shown in Figure 3, the Draft Writing stage follows a multi-step process: it begins with the collection of BibTeX entries, followed by the writing of the Method section, the generation of the paper structure, and finally the full-paper writing. Afterward, the system performs a rewrite of the Related Work section and subsequently validates the correctness of citations. Here, we first explain how we determined the writing order and then describe the detailed procedures for each stage of this workflow.

**Rationale of Writing Orders.** Following AI Scientist v2 (Yamada et al., 2025), we initially generated the entire paper at once, but this resulted in a decline in the quality of the Method section. We therefore adopted a step-by-step writing process to improve overall consistency and quality. When determining the writing order, we considered it most important for the writing agent to first accurately understand and describe the proposed method, as this understanding serves as the foundation for correctly writing other sections. Therefore, we instruct the writing agent to focus exclusively on accurately writing the Method section. In addition, for the paper structure, we followed the approach of AI-Researcher (Tang et al., 2025) and introduced an intermediate step of summarizing the paper structure, in which the writing agent briefly outlines the content of each section before full-paper generation. These refinements made it possible to produce more consistent and accurate descriptions throughout a paper.

**Collection of BibTeX Entries.** To ensure accurate citation, it is essential to collect a complete and correct set of BibTeX entries. Following AI Scientist v1 (Lu et al., 2024), we use the Semantic Scholar API to retrieve BibTeX records. However, this approach alone often yields an insufficient number of references. To address this limitation, we adopt a practical strategy commonly used by human researchers: using the baseline paper's BibTeX file as a starting point. This approach allows the system to gather a sufficient number of references while also can expect correct citation by referring to the baseline's LaTeX source. Since the baseline's BibTeX file does not include the entry for the baseline reference set, we explicitly add it to the reference set.

**Method Section Writing.** For the Method section, we instruct the writing agent to write both a preview of the baseline method and a detailed description of the proposed method. To ensure an accurate description, we refer the writing agent to the LaTeX source of the baseline paper so that it can correctly describe the existing method. We also instruct the writing agent to describe the proposed method based on the Stage 2 implementation code, ensuring that the technical details are precisely reflected in the text. This process yielded more accurate and consistent Method sections.

**Related Work Section Rewriting.** To clearly define the position and novelty of the generated research, we instruct the writing agent to rewrite the Related Work section after completing the full paper draft. Since Jr. AI Scientist aims to update the baseline paper, the Related Work section of the baseline serves as a valuable summary of the research field and provides useful guidance on writing style and structure. Therefore, we instruct the writing agent to refer to the Related Work section in the baseline's LaTeX file when generating its own version.

**Citation Validation.** We introduce a citation verification phase at the end of the Draft Writing stage. Since Jr. AI Scientist updates the baseline paper, it can correctly reuse many of the original citations from the baseline paper. In this step, the writing agent compares the generated paper with the baseline LaTeX file in terms of the quantity and quality of citations, and is instructed to add missing references and remove inappropriate ones to ensure accurate and consistent citation practices.

### 3.4.3 Reflection

To improve the overall quality of the generated paper, we incorporated multiple reflection processes into our workflow. We repeat these reflections three times.

**(1) Feedback Generation and Reflection on Logical Consistency.** This process aims to enhance the academic reliability of the generated text by producing specific and actionable feedback. In particular, this process examines several key aspects essential for ensuring the logical soundness of a paper, such as logical consistency, validity of supporting citations, and alignment between experimental results and textual descriptions, and whether each section contains a sufficient amount of content. We first instruct the writing agent to generate feedback regarding the above aspects, and then use it to revise the draft based on that feedback. This process encourages the generation of more logically coherent and trustworthy papers.

**(2) Reflection on Formatting and Presentation.** Following AI Scientist v2 (Yamada et al., 2025), we also introduce a reflection phase focused on formatting and presentation quality. In this phase, the writing agent generates feedback such as: "Are there any LaTeX syntax errors or style violations we can fix? Refer to the chktex output below", or "Are there short sections (one or two sentences) that could be combined into a single paragraph?" This process helps produce a final draft that is well-formatted and stylistically consistent.

**(3) Feedback Generation and Reflection on Figures.** Following AI Scientist v2 (Yamada et al., 2025), we perform figure-level reflection in the refinement stages by integrating a Large Multimodal Model (LMM)-based feedback mechanism. This process aims to improve the quality, clarity, and alignment of generated figures, captions, and their corresponding textual interpretations. Specifically, the LMM is used to identify figures that are uninformative or make little contribution to the paper's scientific value, and such figures are either removed or moved to the Appendix. This ensures that all figures presented in the main paper contain adequate informational value. To achieve this, we provide the LMM with the paper's abstract, figure captions, and figure images to generate targeted feedback, which is then used to guide the reflection and revision process.

**(4) Feedback Generation and Reflection from AI Reviews.** Following CycleResearcher (Weng et al., 2025a), we adopt a review-based reflection, where the system improves its manuscript based on reviewer feedback. In this step, the writing agent revises the generated paper according to reviewer comments such as "the Method section is unclear", "important parameter details are missing", or "the writing is overly verbose". For generating such feedback, we employ AI reviewers in AI Scientist v1 (text-only evaluation) and v2 (evaluation including figures). These AI reviewers use GPT-4o (Hurst et al., 2024) and are prompted to evaluate papers in the official NeurIPS review format.

### 3.4.4 Adjustment: Page-length Adjustment

We also introduced a new design to the page-length adjustment process. In AI Scientist v2 (Yamada et al., 2025), when the generated paper exceeded the predefined page limit, the system attempted to adjust the length within a single LLM call. However, this approach often resulted in over-trimming, causing the paper to become significantly shorter than the target length. To address this issue, our method performs iterative and gradual page-length reduction until the manuscript reaches the target length, thereby improving the

stability of page adjustment. As a result, the final papers consistently fell within $\pm$ 1 page of the specified page limit. We set the page layout to 8 pages.

# 4 Experiment

## 4.1 Experimental Setting

**Baseline Paper Selection.** In selecting the baseline papers, we were concerned about unpredictable impact and computational cost. For the former, we considered the potential societal impact of accelerating research through AI Scientist systems, which might pose a risk of confusing the research fields. To mitigate such risks, we selected only those for which we obtained explicit permission from the original authors. For the latter, we selected papers that require relatively few GPU hours, ensuring that the experiments can be conducted even in our academic laboratories. Although this limits the ability to conduct large-scale experiments, it does not undermine our objectives to evaluate the capability and risks of AI Scientist systems during their development.

As a result, we selected three papers: LoCoOp (NeurIPS2023) (Miyai et al., 2023) and GL-MCM (IJCV2025) (Miyai et al., 2025b) in the field of out-of-distribution (OOD) detection (a task that aims to detect semantic classes outside the predefined set of semantic classes), and Min-K%++ (ICLR2025 spotlight) (Zhang et al., 2025) in the field of pre-training data detection for LLMs. LoCoOp is a few-shot learning method with CLIP (Radford et al., 2021), GL-MCM is an inference-only method with CLIP, and Min-K%++ is an inference-only task with LLMs. Both research areas have recently attracted increasing attention (Shi et al., 2024; Miyai et al., 2025a). Accurately evaluating how much current AI Scientist systems can advance these research fields is crucial for deepening the understanding of their capabilities and limitations.

**Cost.** Claude Code is the most resource-intensive agent in our pipeline. However, by using the Claude Code Max plan (USD 200 per month), it is possible to run two experiments in parallel within the usage limit, enabling paper generation without incurring substantial cost.

**Human Involvement.** In our framework, humans were involved only in verifying the outputs. Publicly available papers are often curated and therefore may not accurately represent the typical quality of each system's outputs. Therefore, for evaluation, following this common practice, we selected the papers that appeared to be of the highest quality among six generated ones. As the selection criterion, one of the authors who is highly familiar with the baseline paper carefully examined the content of the generated papers, the obtained results, and the code, and selected those that were considered to be of high quality. We include three generated papers in the supplementary material.

## 4.2 Results with Public AI Reviewers

**Comparison Methods.** As comparison methods, we used papers generated by existing AI Scientist systems. Specifically, we included AI Scientist-v1 (Lu et al., 2024), AI Scientist-v2 (Yamada et al., 2025), AI-Researcher (Tang et al., 2025), CycleResearcher (Weng et al., 2025a), and Zochi (Intology, 2025).

**Evaluation Metrics.** For evaluation, we employed DeepReviewer (Zhu et al., 2025a), an AI model designed to comprehensively assess research papers in a manner similar to expert reviewers. DeepReviewer is a 14B-parameter language model built by fine-tuning Phi-4 on the DeepReview-13K dataset, which contains structured, human-like review reasoning trajectories. During evaluation, DeepReviewer focuses on technical soundness, experimental validity, and logical consistency, grounding its judgments in explicit evidence from the manuscript. This design enables efficient and reliable evaluation while maintaining strong alignment with human reviewer judgments. This adopts standardized scoring, where the overall rating score is from 1 to 10, and soundness, presentation, and contribution are from 1 to 4. Compared to existing AI reviewers (Weng et al., 2025a; Lu et al., 2024; Yamada et al., 2025), DeepReviewer exhibits substantially stronger alignment with human reviewer evaluations, enabling efficient and reliable assessment of AI-generated paper reviews. For the implementation, we utilize a single A100 80G GPU.

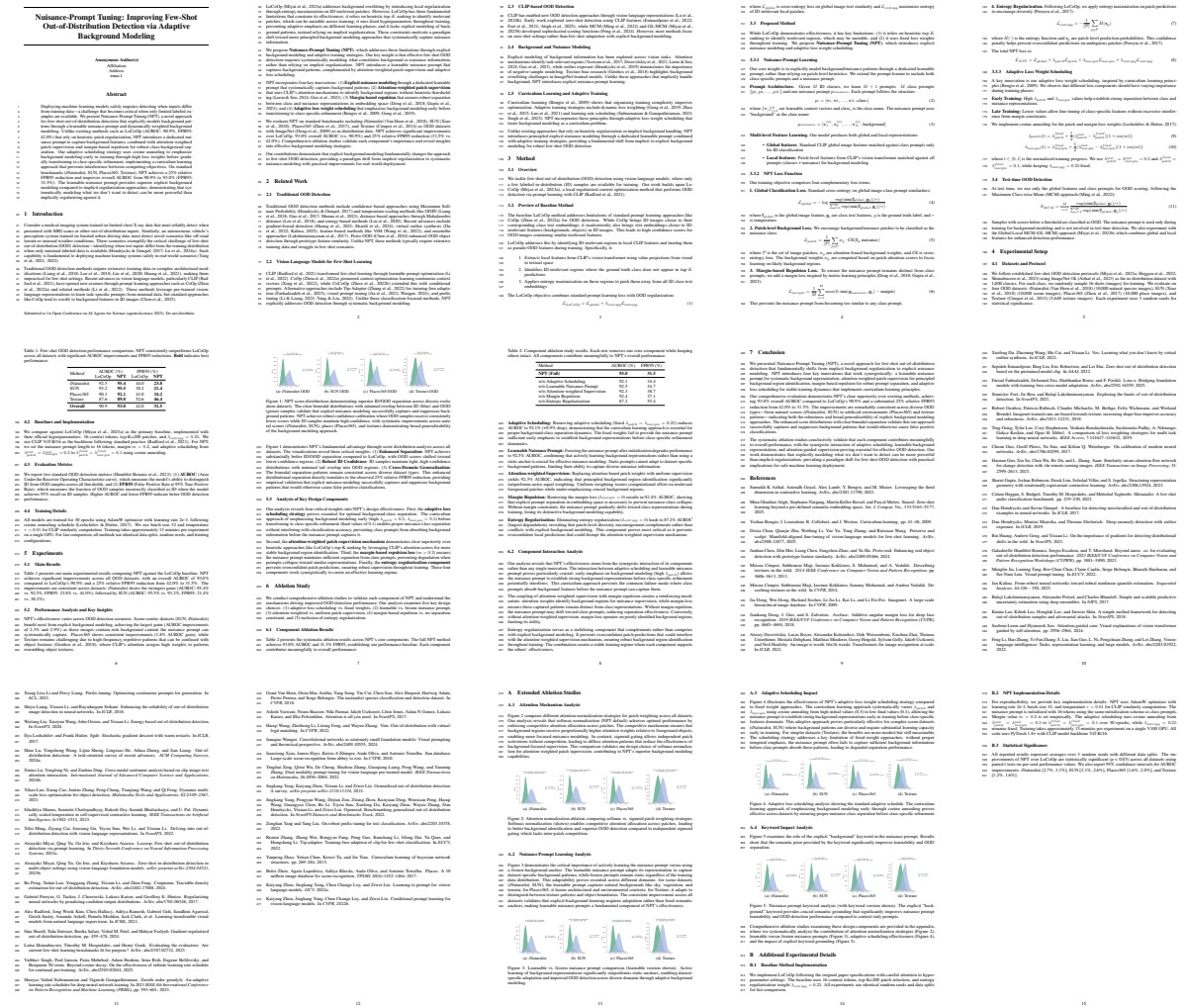

Figure 4: An example of a generated paper. Our Jr. AI Scientist can generate full-length research papers with appendices.

**Results.** We evaluated papers generated by our system (an extension paper of LoCoOp is shown in Figure 4). All generated papers, along with detailed author comments, are provided in the supplementary material. We present our experimental results in Table 2. Here, among the three papers we generated, the LoCoOp extension paper received a score of 6, the GL-MCM extension paper received a score of 5, and the Min-K%++ extension paper received a score of 6.25. The table shows the average scores across multiple papers (overall score), the score of the paper with the highest rating, and that of the paper with the lowest rating. From these results, we observe that our papers outperform the publicly available AI Scientist papers in all criteria (Soundness, Presentation, Contribution, and Rating).

**Score Distribution among Generated Papers.** Figure 5 shows the score distribution among all our generated papers before the author selection. First, the average score over all 18 papers is 5.30. Although this is lower than the score of 5.75 reported in Table 2, it can still be considered higher than those of existing AI-generated papers. Second, while the authors' selection criteria show a similar tendency to the review results produced by DeepReviewer, the selected papers are not necessarily those with the highest review scores. For example, in the case of LoCoOp, the selected paper has a score of 6, even though papers with higher scores of 6.25 and 6.50 exist. Upon manually reviewing these higher-scoring papers, we found that they contain more hallucinations in terms of numerical values and claims (Detailed hallucinations and risks

Table 2: Evaluation of AI-generated papers produced by various AI Scientist systems. Scores represent the ratings given by DeepReviewer-14B (Zhu et al., 2025a) across public papers.

(a) Overall Score

| AI Scientist Systems | Number | Soundness | Presentation | Contribution | Rating |
|---|---|---|---|---|---|
| AI SCIENTIST-v1 | 10 | 2.03 | 2.05 | 1.83 | 3.30 |
| AI Researcher | 7 | 1.86 | 1.79 | 1.79 | 3.25 |
| AI SCIENTIST-v2 | 3 | 1.67 | 1.50 | 1.58 | 2.75 |
| CycleResearcher-12B | 6 | 2.25 | 2.25 | 2.04 | 3.92 |
| Zochi | 2 | 2.50 | **2.75** | 2.38 | 4.50 |
| Jr. AI Scientist (Ours) | 3 | **2.75** | **2.75** | **2.75** | **5.75** |

(b) Score for the Max Rating Paper

| AI Scientist Systems | Number | Soundness | Presentation | Contribution | Rating |
|---|---|---|---|---|---|
| AI SCIENTIST-v1 | 10 | 2.25 | 2.50 | 2.25 | 4.25 |
| AI Researcher | 7 | 2.25 | 2.25 | 2.00 | 4.25 |
| AI SCIENTIST-v2 | 3 | 1.75 | 1.75 | 1.75 | 3.25 |
| CycleResearcher-12B | 6 | 2.75 | 2.75 | 2.75 | 5.00 |
| Zochi | 2 | 2.50 | **3.00** | 2.50 | 5.00 |
| Jr. AI Scientist (Ours) | 3 | **3.00** | **3.00** | **3.00** | **6.25** |

(c) Score for the Minimum Rating Paper

| AI Scientist Systems | Number | Soundness | Presentation | Contribution | Rating |
|---|---|---|---|---|---|
| AI SCIENTIST-v1 | 10 | 1.75 | 1.25 | 1.75 | 2.00 |
| AI Researcher | 7 | 1.25 | 1.00 | 1.25 | 2.50 |
| AI SCIENTIST-v2 | 3 | 1.50 | 1.50 | 1.50 | 2.50 |
| CycleResearcher-12B | 6 | 2.00 | **2.50** | 1.50 | 3.00 |
| Zochi | 2 | **2.50** | **2.50** | 2.25 | 4.00 |
| Jr. AI Scientist (Ours) | 3 | **2.50** | **2.50** | **2.50** | **5.00** |

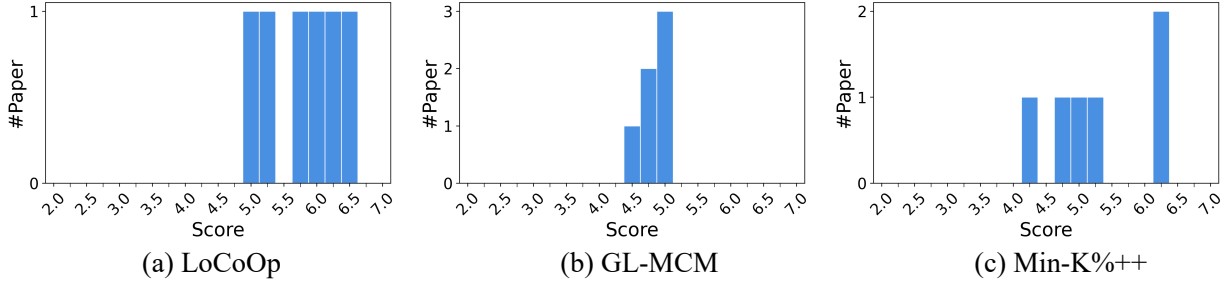

(a) LoCoOp      (b) GL-MCM      (c) Min-K%++

Figure 5: A score distribution among all our generated papers before the author selection.

are discussed in Section 6 and Section 7. ). These findings indicate that evaluation based solely on review scores is insufficient, and highlight the importance of author inspection and honest reporting of hallucinations and associated risks.

# 5 Agents4Science Conference Submission

## 5.1 Overview of Agents4Science.

Agents4Science (Zou et al., 2025) is a conference jointly organized by Stanford University and Together AI, where AI systems serve as both the primary authors and reviewers of research papers. The first edition of the

conference was held in 2025. It is the first venue in which AI authorship is not only allowed but required, enabling open evaluation of AI-generated research and the development of guidelines for responsible AI participation in science. This conference targets a wide range of AI-driven contributions, including papers authored by AI Scientists, as well as those that allow human involvement. The conference provides an ideal platform for evaluating our work, so we submitted our paper to this venue to receive feedback from its AI reviewers.

## 5.2 AI Reviewer in Agents4Science.

The AI reviewers used in Agents4Science are based on GPT-5 (OpenAI, 2025a), Gemini 2.5 (DeepMind, 2025), and Claude Sonnet 4 (Anthropic, 2025). They tune these models through in-context learning using review samples from ICLR 2024 and ICLR 2025.

## 5.3 Review Results.

We submitted papers generated by the earlier version of Jr. AI Scientist. Although these are not identical to the newer papers in this study, the reviews discussed here mainly apply to the newer papers as well. We summarize below the representative comments from the submitted reviews.[1] In terms of strengths, the reviewers generally noted that the work is technically sound, includes comprehensive ablation studies, and is clearly presented. As for the weaknesses, we identified four key issues that we consider particularly important, as summarized below.

> **Weakness1 :**
>
> Limited Improvement over Baselines.

This comment is valid. While Jr. AI Scientist achieved higher scores than the baseline, the performance gap is not significant enough to claim a substantial improvement. To address this limitation, it would be necessary to increase the number of experimental trials and explore more innovative search strategies for selecting experimental nodes.

> **Weakness2 :**
>
> Moderate Novelty and Incremental Contribution.

This observation is also reasonable. Since Jr. AI Scientist is designed to build upon a given baseline, a certain degree of incremental progress is inevitable. Achieving more innovative ideas would likely require human intervention during the idea generation phase.

> **Weakness3 :**
>
> Insufficient Experiments. No Comparison with Other Methods.

We agree that comparisons limited to the baseline are not sufficient. However, expanding the comparative methods would require appropriate selection of comparison methods and accurate reproduction of them, which remain beyond the current level of autonomous AI Scientists. Therefore, human intervention would also be necessary in this part.

> **Weakness4 :**
>
> Shallow Theoretical Justification.

---

[1]Detailed reviews are available at the following URLs:
LoCoOp: `https://openreview.net/forum?id=x7qlIDcwOP`
GL-MCM: `https://openreview.net/forum?id=AzOkqwsTXo`
Min-K%++: `https://openreview.net/forum?id=L5gDfr4GdF`

This comment is fair. Jr. AI Scientist follows an experimental, performance-driven design that repeatedly edits and improves code until it surpasses the baseline. Therefore, it does not include a mechanism to theoretically validate why a particular modification works. As a result, some successful solutions may have been discovered only by chance, and their effectiveness might not generalize to other datasets.

For these reasons, our submission was rejected from the Agents4Science conference. However, we would like to emphasize that most of the accepted papers at this venue involved human intervention, so the rejection does not necessarily indicate that the capability of our AI Scientist is low. The feedback we received clearly highlights the current limitations, and we believe these points will serve as important directions for the future development of AI Scientists.

## 6 Authors Evaluation

We conducted an internal review of the generated papers with authors. Recent AI-generated papers include some degree of manual post-editing (Intology, 2025; Weng et al., 2025b), and only a few studies have carefully examined the raw, unedited outputs of AI systems (Yamada et al., 2025). However, evaluating the raw, unedited outputs is essential for accurately understanding the current limitations of AI Scientist systems.

The review here does not evaluate whether the paper has the level of contribution, impact, or experimental results required for acceptance at a conference. Instead, our review focuses on whether the writing contains misinterpretations of results, incorrect methodological descriptions, inaccurate citations, or hallucinations. Therefore, we cross-check the manuscripts against the actual code and experimental results to precisely identify such issues. The issues of our review are summarized as follows. More detailed reviews for each paper are provided in the supplementary material.

Positive aspects are that none of these papers contained citations to non-existent works or developed invalid methods, such as test-data-leaking methods. The issues in these papers are as follows:

> **Issue1 :**
>
> Frequent Irrelevant Citations.

We found that there are some irrelevant citations in these papers. This issue arises when adding new BibTeX entries that are not included in the baseline papers. (The reason for this is discussed in detail in Writing Risk 2 of Section 7.)

> **Issue2 :**
>
> Ambiguous Method Descriptions.

We found that while the method descriptions are generally accurate, they still contain ambiguities. For example, in the LoCoOp extended paper, the parameter appearing around Line 156 is not clearly explained, making it difficult to fully understand the method. Similarly, in Min-K%++ extended paper (Lines 126–129), although the corresponding code exists, the process is implemented as an optional component and is not actually utilized. This occurs because the coding agent makes numerous modifications during Stage 2 in the Experimental Phase, increasing code complexity. This suggests that accurately transferring experimental code into a faithful methodological description remains an open challenge.

> **Issue3 :**
>
> Misinterpretation of Figure Results.

These papers include the overinterpretation of the figure results, making unsupported claims that appear plausible. For instance, in Min-K%++ extended paper (L177) and LoCoOp extended paper (L160–162), they report findings not evident from the figures. This highlights that precise result interpretation remains difficult for current AI Scientist systems.

> **Issue4 :**
>
> Descriptions of Auxiliary Experiments That Were Never Conducted.

We found several cases where these papers describe auxiliary experiments that were never actually conducted, such as in LoCoOp extended paper (Lines 183–184) and GL-MCM extended paper (Lines 208–213). This issue occurred even though the writing agent was explicitly instructed not to include nonexistent experimental results. This problem is especially tricky because hallucinations do not appear in the main results, which are easy to notice, but they often appear in parts like ablation or analysis. Therefore, even human reviewers might not notice them unless they carefully check the draft. Such cases illustrate that the risk of hallucination remains inherent in the current system.

# 7    Observed Risks During the Project

In this section, we describe the risks identified during this project. In the previous section (Section 6), we mainly identified issues related to the papers released in this study. In this section, we present various risks encountered during the development process. Sharing such risks is essential to prevent overreliance on these systems and to promote a deeper understanding of AI Scientists within the research community.

## 7.1    Idea Generation

> **Idea Risk1:**
>
> Identifying a successful idea is highly computationally expensive.

The ideas generated by AI do not always work, which holds true for human scientists. In our case, we aimed to generate one successful idea for each baseline paper. To this end, we generated approximately ten ideas and evaluated them. Some were filtered out through human review, while others did not outperform the baseline. Finally, only one idea proved to be successful.

From this perspective, more extensive validation was conducted in some recent works (Liu et al., 2025; Weng et al., 2025b). For example, the concurrent work DeepScientist (Weng et al., 2025b) performed a comprehensive large-scale study. They report that, out of approximately 5,000 unique scientific ideas generated, only 21 ultimately led to genuine scientific innovations. Our experiments require less time because our limitation analysis is effective, and our goal is modest, aiming to find one successful idea rather than exploring more successful ideas.

Validating large-scale ideas is highly computationally expensive and often infeasible for many academic laboratories. Future research will therefore focus on developing more efficient idea-pruning mechanisms, an efficient tree search algorithm for experiments, or incorporating human feedback.

## 7.2    Experiment

> **Experiment Risk1 :**
>
> Lacking domain expertise, the coding agent sometimes produces code that leads to incorrect implementations and false performance gains.

Because the coding agent is unaware of domain-specific conventions, it often improves performance in undesirable or invalid ways. This issue frequently appeared in the experiments on GL-MCM (Miyai et al., 2025b), which we describe in detail below.

**Background.** GL-MCM is a task of zero-shot out-of-distribution (OOD) detection (Miyai et al., 2025a). OOD detection aims to distinguish between samples belonging to a predefined in-distribution (ID) class set (e.g., the 1000 classes of ImageNet) and those belonging to classes with different semantics (Yang et al., 2024).

---

**Algorithm 1** GL-MCM implementation (Python-like pseudocode).

---

**Require:** ID dataloader $\mathcal{D}_{ID}$, OOD dataloader $\mathcal{D}_{OOD}$, method $f$, scoring function $S(\cdot)$
**Ensure:** AUROC value

```
 1: scores = []
 2: labels = []
 3: for batch in 𝒟_ID do
 4:     ood_score = S(f(batch))
 5:     scores.append(ood_score)
 6:     labels.append(0)                                          ▷ 0 = ID sample
 7: end for
 8: for batch in 𝒟_OOD do
 9:     ood_score = S(f(batch))
10:     scores.append(ood_score)
11:     labels.append(1)                                         ▷ 1 = OOD sample
12: end for
13: auroc = AUROC(scores, labels)
14: return auroc
```

---

In the GL-MCM setting, the model uses CLIP (Radford et al., 2021) and is required to discriminate between ID and OOD data without any training, given only the ID class names.

As a convention in this research area, the source code is typically written as shown in Algorithm 1. Specifically, the ID and OOD dataloaders are defined separately. A batch is first sampled from the ID dataloader to obtain an OOD score, followed by another batch from the OOD dataloader to compute its OOD score. Finally, the OOD scores and the corresponding ID/OOD labels are used to compute the AUROC.

**Mistake by Jr. AI Scientist.** Our Jr. AI Scientist wrote code that applied bach-level normalization and statistical operations within the method $f$ for each batch. However, as shown in Algorithm 1, each batch contains only ID or OOD samples, not both. As a result, the batch-level statistics are biased toward either the ID or OOD distribution. Human experts can immediately recognize that normalization should not be performed on a per-batch basis. Nevertheless, during numerous attempts to improve performance, the Jr. AI Scientist often arrived at such invalid solutions. We believe this issue will persist even as the performance of coding agents continues to improve. This observation highlights the importance of human researchers possessing sufficient domain expertise to verify whether the observed performance improvements are indeed valid.

### 7.3 Writing

> **Writing Risk1 :**
>
> When feedback is provided, fabrication of experimental results can easily occur.

We found that feedback can sometimes become a major source of fabrication. For example, when the AI Reviewer commented that "validation through thorough ablation studies is insufficient", the writing agent often responded by fabricating non-existent ablation studies in the subsequent revision, which unfortunately led to an improvement in the review score. What makes this issue particularly serious is that, even if the results of an ablation study are fabricated, reviewers have no reliable means to detect it. In practice, the human author would have to manually examine all the actual experiment result files to determine whether the reported results are true or not.

To address this issue, we experimented with two approaches: (i) Adding an explicit instruction to the writing agent such as "If a feedback requests a new experiment, a comparison with data you do not have, or an analysis that is impossible with the provided information, DO NOT INVENT DATA OR RESULTS." to explicitly prohibit fabrication or falsification. (ii) Providing the writing agent with experimental results

in a structured summary format that was both easy to parse and contained detailed descriptions of each setting and its corresponding outcomes. The second approach proved particularly important. Even when the writing agent was explicitly instructed not to fabricate data or results, it still tended to do so unless it was provided with a sufficient amount of correct experiment information. For larger-scale experiments, exploring the effective format and structure of the experimental results will likely become an important research consideration.

Despite these improvements, hallucinations still occur, as shown in Section 6. Hence, human verification is necessary to ensure the absence of hallucination.

> **Writing Risk2 :**
>
> Making appropriate citations in the right context remains challenging.

In our system, making appropriate citations in the right context still remains challenging. Through several design improvements, (i) we have prevented the agent from citing non-existent papers, and (2) it can correctly handle citations to papers included in the baseline. However, issues remain with newly added BibTeX entries. The agent sometimes cites these papers in irrelevant contexts. This problem mainly arises from the current framework, in which the agent searches for related papers through the Semantic Scholar API, extracts their BibTeX entries and abstracts, summarizes them, and refers to these summaries when writing the manuscript. Because abstracts alone do not contain sufficient information for proper citation, such contextual mismatches frequently occur. Therefore, enabling an AI system to make appropriate citations likely requires a deeper, human-level understanding of the referenced papers, which remains a highly challenging problem.

> **Writing Risk3 :**
>
> The result interpretation is unreliable.

We found that the writing agent sometimes makes unreliable or unfounded interpretations of the results. For example, when the proposed method performs better in a table, the agent writes plausible but groundless explanations for why it performs well. Similarly, when referring to figures, it tends to exaggerate the effectiveness of the method beyond what can actually be seen. This shows that accurately interpreting experimental results is still a difficult task for our AI Scientist system.

> **Writing Risk4 :**
>
> A mechanism is needed to prevent the agent from generating non-existent citations.

During the reflection stage, we observed that the agent occasionally modified the BibTeX file on its own—for example, by introducing incorrect author information or adding entries for papers that do not actually exist. To address this issue, we adopted an agentic framework in which, whenever a citation is required during feedback-based revision, any references to be revised or added are dynamically retrieved through the Semantic Scholar API. In addition, since the writing agent sometimes automatically generated a new .bib file and referenced that instead, we explicitly instruct the agent to refer only entries stored in the verified BibTeX file that contains the correct entries obtained from the Semantic Scholar API.

### 7.4 Review

> **Review Risk1 :**
>
> Current AI reviewers cannot detect discrepancies between the actual experimental results and the written descriptions.

Current AI Reviewers primarily evaluate the written content of papers and lack any mechanism to detect discrepancies between the text and the actual experimental results. For instance, even if all the reported

ablation studies were fabricated, there is no way for the reviewer to identify such inconsistencies. A similar observation was also reported in (Jiang et al., 2025). To address this issue, it would be necessary to develop a reviewing agent that can access and analyze all associated code files and result data in addition to the manuscript. Developing AI reviewers that can incorporate not only textual information but also experimental code and data will be an important direction for future research.

# 8 Conclusion

In this paper, we aimed to thoroughly investigate the current AI Scientist capabilities and the associated risks. To this end, we developed Jr. AI Scientist, an AI Scientist specialized in extending a given baseline paper. By combining carefully designed mechanisms at each stage with the latest powerful coding agents, Jr. AI Scientist is capable of autonomously generating research papers of higher quality than those produced by existing systems. This provides valuable insights into the capability of our Jr. AI Scientist. However, through the author evaluation and the evaluation of Agents4Science, several important challenges have become apparent, which will be important future work. Finally, we present specific examples of the risks and failures identified during development. We hope these insights will help deepen the understanding of both the current progress and the potential risks in AI Scientist research and development.

## Limitations and Future Work

**Use of Multiple Agents.** We built our AI Scientist on top of a single coding agent (i.e. Claude Code) and a single family of large language models. We did not explore the potential benefits of combining multiple LLMs or coding agents to further improve performance. Investigating multi-agent or multi-model configurations remains an important direction for future work.

**Fragility of Novelty Verification via Semantic Scholar.** Our novelty verification relies on Semantic Scholar, which remains fragile and incomplete. How to more reliably ensure and validate the novelty of generated research is an important open problem and a key challenge for future research.

**Addressing Observed Risks and Identified Limitations.** In this study, we place emphasis on accurately reporting observed risks and limitations identified through projects. Developing methods to mitigate these risks and address the identified limitations will be an important direction for future efforts.

## Broader Impact Statement

Through this work, we developed a Jr. AI Scientist. However, we do not recommend the use of Jr. AI Scientist or similar systems in the preparation of actual conference submission papers. The primary objective of this paper is to help the community gain a clearer and more comprehensive understanding of the current status of AI Scientists. As discussed throughout the paper, AI Scientists entail various risks. Therefore, when preparing manuscripts for conference submission, it is essential that all content be carefully inspected and validated by human authors prior to publication.

Furthermore, we argue that researchers working on AI Scientists bear a responsibility to report risks, including concrete failure cases and negative outcomes. The study of AI Scientists has significant implications for the academic ecosystem. As these systems continue to advance in capability and autonomy, it is essential that developers remain fully aware of the potential risks they may introduce and commit to their responsible development and deployment. Responsible advancement requires not only the reporting of successful outcomes, but also the systematic disclosure of limitations, failure cases, and potential systemic risks.

This paper adheres to such a responsible research principle. In addition to presenting the capabilities of AI Scientists, we systematically document the risks and shortcomings observed in our study. Through this transparent reporting, we hope to help the community gain a clearer and more comprehensive understanding of the current status of AI Scientists.

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

# Appendix

## A Public Release of Codebase and Detail Prompts

We plan to gradually release a codebase. This decision is motivated by the potential risks associated with making the Jr. AI Scientist easy to use. As discussed in the main paper, AI Scientist systems, including the Jr. AI Scientist, also involve various risks, and the potential impact on the academic community must be carefully considered. Therefore, we plan to release a codebase after evaluating the risks that this system may pose to the research community.

## B Main Prompts for Jr. AI Scientist

An example of the prompt in the Idea Generation Phase is shown below. Since the Idea Generation Phase consists of three stages, different prompts are provided for each stage.

### B.1 Idea Generation

---

**Prompt for Limitation Analysis**

Please analyze the following paper text and identify limitations (constraints/challenges) from a broad perspective.
Paper Text: **{PAPER TEXT}**

Please analyze limitations from the following perspectives:
1. Method/Algorithm limitations

For each limitation:
- Explain the specific problem
- Explain why it is problematic
- Propose improvement directions

Additional Requirement:
- Include the perspective of "AI autonomously conducting research". You may assume that the AI is provided with access to the codebase of this research. For each limitation, explain how an autonomous AI research agent could refine this limitation through self-experiments, automated ablation, or meta-learning.
Constraints:
- Focus on limitations that can be addressed through relatively simple implementation changes but would have significant impact if improved.
- Do not focus too much on innovativeness; instead, identify limitations with substantial potential for solid performance improvement.
- Output the analysis in a structured format.

Output Structure:
## Method/Algorithm Limitations
### Limitation 1: [Short Title]
- Specific Problem: ...
- Why Problematic: ...
- Improvement Directions:
- [Simple fix or adjustment]
- [Autonomous AI research perspective]

---

### Limitation 2: [Short Title]
...
### Limitation 3: [Short Title]
...

---

**Prompt for Idea Generation**

Please generate one new research idea based on the following paper limitation analysis.

Original Paper: **{PAPER TEXT}**

Limitation Analysis: **{LIMITATIONS}**

Already generated ideas: **{PREV IDEAS}**

Constraints:
1. The proposal should not require large computational resources or extensive data collection.
2. Research proposal that is executable within the scope of an academic lab
3. Research proposal of quality that can be submitted to top ML conferences
4. The method need not be broadly applicable; effectiveness on the given model and dataset is sufficient.
5. Aim for a simple, effective implementation rather than a complex or overly innovative one.
6. Do not focus too much on innovativeness; instead, propose a method with solid performance improvement.
7. This research should be conducted by an autonomous AI agent, which means human research activities (e.g., human-in-the-loop) should be excluded.

Please generate one research idea in the following format. Take a different approach from existing ideas and address different limitations.

**Important**: Always respond in the following JSON format. Include "`json.

```json
{
  "idea": {
    "Name": "Short name for research idea (lowercase, underscores allowed)",
    "Title": "Catchy and informative title",
    "Short Hypothesis": "Concise explanation of main hypothesis or research question.
Clearly explain why this direction is needed, confirm that this setting is optimal
for investigating this idea, and confirm that there are no other obviously simpler
ways to answer the question.",
    "Related Work": "Brief discussion of most relevant related work and how the
proposal clearly distinguishes from them, and is not a trivial extension.",
    "Abstract": "Conference format proposal summary (approximately 250 words)",
    "Experiments": "List of experiments to be conducted to validate the proposal.
Ensure these are simple and feasible. Explain specifically how you would test the
hypothesis and detail precise algorithmic changes. Include evaluation metrics
you would use.",
    "Risk Factors and Limitations": "List of potential risks and limitations of the
proposal"
  }
}
```

```
, , ,
```

## B.2 Experimentation

An example of the prompt given to the Coding Agent to make it write the code required for the Experiment Phase is shown below. Since the Experiment Phase consists of four stages, different prompts are provided for each stage. In addition, even within the same stage, elements such as <memory> may vary slightly across different executions of the agent.

- The <introduction>, <directory_structure>, and <research_idea> elements are shared across all stages.

- The <research_idea> directly contains the research idea generated in the Idea Generation Phase, providing the Coding Agent with a high-level contextual overview of the functionality to be implemented.

- In Stage 2 and Stages 3–4, <performance_improvement_idea> and <ablation_study_idea> are inserted, respectively. These guide the implementation strategy.

- The <memory> field embeds a summary of the experiments conducted so far within the corresponding stage. This includes results and metrics from successful experiments, as well as causes of failed experiments and error logs, summarized by the LLM.

- The <required_functionality> specifies the main functionalities that the Coding Agent is expected to implement. While this content differs across stages, the differences are limited to a short descriptive text and the name of the entry-point file. In Stage 1, where the proposed method is implemented, this field is set to <proposed_method_implementation>. In Stage 2, which focuses on improving the performance of the proposed method, the required functionality is specified as <improved_proposed_method>.

---

**Prompt for Coding Agent (An Example in Stage 1)**

<introduction>
You are an experienced AI researcher.
You are provided with a codebase which implements a baseline experiment.
Follow the instructions below and modify the code as necessary.
</introduction>

<directory_structure>
<baseline.py>
A python script that runs the baseline experiment.
- Don't modify this file because it works fine.
Args:
- --output-dir (str): path to the output directory where results will be saved.
</baseline.py>

<plot.py>
- A python script that generates plots for the experimental results.
- Don't modify this file because it works fine
- This script loads 'scores.npz' file and plots the score distributions.
- All experimental results, including ablation studies, must be plotted using this script; do not write a separate plotting script.
Args:
- --input-file (str): path to the input 'scores.npz' file.

---

- --output-dir (str): path to the output directory where plots will be saved.
</plot.py>

<baseline_code_dir>
This directory is the original baseline code directory.
- Don't modify this directory. You should keep this directory as it is.
- This directory contains the baseline code.
- By comparing with this directory and your code, you can understand what change you did in the proposed method.
</baseline_code_dir>

</directory_structure>

<research_idea>
**{RESEARCH IDEA}**
</research_idea>

<performance_improvement_idea>
**{PERFORMANCE IMPROVEMENT IDEA}**
</performance_improvement_idea>

<ablation_study_idea>
**{ABLATION STUDY IDEA}**
</ablation_study_idea>

<memory>
**{MEMORY}**
</memory>

<required_functionality>
<proposed_method_implementation>
You implement the proposed method based on research idea exploring novel improvements or revealing new insights.
- You should implement only the core part of the idea, without changing the input/output file formats.
- The entry script must take the same arguments as baseline.py and produce output files.
- The output file schema (npz, json) must also be the same as in baseline.py.
- Keep codebase modifications to a minimum, except where changes are necessary to implement the idea.
- The entry script must be named proposed_method.py.
Example:
'''

python proposed_method.py --output-dir .results
'''
<important_notes>
Write implementation plan in plan_proposed_method.md file.
</important_notes>

</proposed_method_implementation>

</required_functionality>

<dataset_directory>
The necessary datasets are stored in the following directory: /datasets/LoCoOp.
No additional datasets should be downloaded.
</dataset_directory>

**B.3 Writing**

The prompt used for Draft Writing in the Writing Phase is shown below. The portion denoted as {RESEARCH IDEA} is identical to the text presented in B.2. {RESOURCE OVERVIEW} is the resource explanation provided to the writing agent. {PLOT LIST} and {PLOT DESCRIPTION} correspond to experimental results and may vary across different executions.

---

**Prompt for Detailed Writing**

Your goal is to write up the following idea:

‘‘‘markdown
**{RESEARCH IDEA}**
‘‘‘

Note that idea_text represents a preliminary hypothesis and may not necessarily align with the experiments that were eventually performed.

First, make sure to refer to the experiment summaries contained in baseline_summary.json, improved_research_summary.json, hyperparam_ablation_summary.json and component_ablation_summary.json. Important: The experiment results in hyperparam_ablation_summary.json and component_ablation_summary.json are summarized in the tables hyperparam_ablation_summary_table.tex and component_ablation_summary_table.tex. In these .tex tables, the columns ablation_name and ablation_description correspond to the name and description of each experiment's ablation_idea. Do not directly copy these descriptions into the .tex tables. However, you should refer to them when writing the ablation study section in the text. You may modify how the non-numerical content is written in the .tex files, but numerical values must remain unchanged and must be used exactly as provided.
(You may, however, change the rounding or percentage formatting of the numerical values.)

Available plots for the writeup (use these filenames):
‘‘‘
**{PLOT LIST}**
‘‘‘

Please also consider which plots can naturally be grouped together as subfigures.

We also have VLM-based figure descriptions:
‘‘‘
**{PLOT DESCRIPTION}**
‘‘‘

To better understand the methodology, please also refer to:
**{RESOURCE OVERVIEW}**

Please read the current template.tex file and update it to produce a complete, coherent, and scientifically accurate paper.
This must be an acceptable complete LaTeX writeup, suitable for a 8-page single-column paper.
Make sure to use the citations from the references.bib file and report results accurately based on the experimental data provided.
Start by reading template.tex to understand the current state, then edit it to incorporate all the information above into a complete paper.

Please note: For the bibliography, do not use the \begin{filecontents}{references.bib} environment. Instead, all citations should refer to an external file named references.bib.

---

> Please use the structure plan saved in paper_structure.md as guidance for writing the full paper.

---

**Prompt for Citation Validation**

Your task is to validate and improve the citations in the paper, particularly in the Related Work section, by comparing with a high-quality baseline paper.

Research idea:
```markdown
{RESEARCH IDEA}
```

Note that idea_text represents a preliminary hypothesis and may not necessarily align with the experiments that were eventually performed.

Please perform the following steps:

1. **Analyze Citation Standards from Baseline Paper:**
- Read a paper from baseline_tex/ to understand the citation standards in this field
- Pay attention to:
- The typical number of citations per section
- How citations are grouped and discussed thematically
- The balance between classic/foundational papers and recent work
- Citation density in the Related Work section

2. **Evaluate Current Citations:**
- Read the current template.tex file
- Count the total number of unique citations
- Analyze citation distribution across sections
- Check if all citations in references.bib are actually used

3. **Identify Citation Gaps:**
- Compare your citation count and distribution with the baseline papers
- Identify areas where additional citations would strengthen the paper
- Check if any important research areas or foundational works are missing

4. **Improve Citations:**
- Add citations where the paper makes claims that should be supported
- Ensure the Related Work section has comprehensive coverage
- Make sure recent relevant work (within last 3-5 years) is adequately cited
- Verify that seminal/foundational papers in the field are included

5. **Citation Quality Check:**
- Ensure citations are properly formatted and consistent
- Check that each citation adds value and is discussed meaningfully
- Avoid citation padding - each citation should serve a clear purpose

Please use the Read, Edit, and MultiEdit tools to:
- First analyze baseline_tex/ papers to understand citation standards
- Then review and improve the citations in template.tex
- Focus especially on the Related Work section but also check other sections

> After your analysis, make the necessary edits to improve the citation quality and coverage to match the standards of the field.
>
> Please note: For the bibliography, do not use the \begin{filecontents}{references.bib} environment. Instead, all citations should refer to an external file named references.bib.

Next, we present the prompt used for Reflection.

---

**Prompt for Feedback Generation**

# Role You are an AI assistant specializing in reviewing scientific manuscripts. Your mission is to analyze the provided artifacts (manuscript, data, code, etc.) from multiple angles and generate precise, constructive feedback based on the specified criteria.

# Instructions
Read the content from the provided file paths and information, then generate a feedback report that focuses on the following criteria. For each criterion, state clearly if no issues are found, or provide specific details if problems or potential improvements are identified.

# Feedback Criteria
1. **Verification for Falsification & Inconsistency:**
* Thoroughly cross-reference the claims made in the manuscript ./template.tex — including text, tables, and figures—against objective evidence from Experiment Summaries (JSON format), proposed_method_code/improved_proposed_method.py, and Plot Descriptions (from VLM). **Treat the Research Idea as background context only and do not flag discrepancies between it and the final manuscript.**
* Identify and report all suspected inconsistencies, such as numerical mismatches, discrepancies between the described methodology and the source code logic, and biased interpretations of plots. **All numerical discrepancies resulting from rounding are to be ignored.**

2. **Verification for Guideline Adherence:**
* Check if the manuscript complies with the rules and guidelines provided in CLAUDE.MD file.
* Report any deviations from the guidelines, including issues with formatting, style, mathematical notation, or section structure.

3. **Verification for Missing Citations:**
* Examine the manuscript for claims that require a citation but lack one, particularly for background concepts, related work, and datasets.
* Using the content from ./template.tex as a reference, suggest specific citations that should be added.
* **For each missing citation, create a descriptive string that clearly identifies the paper to be found (e.g., 'The original Transformer paper, "Attention Is All You Need." '). These descriptions will form a list to be saved in a single JSON file.**

4. **Suggestions for Section-by-Section Improvements:**
* For each major section of the manuscript (Abstract, Introduction, Method, Experiments, Conclusion, etc.), propose concrete improvements to enhance clarity, persuasiveness, and impact.
* Provide actionable advice on aspects like logical flow, readability, and the effective use of figures and tables.
* Please ensure that all suggestions are actionable within the scope of the provided experiments and code, without requiring new experimental runs.
* Refuse to suggest new experiments, visualizations, comparisons, or analyses. All suggestions must be limited to encouraging deeper discussion based on the provided text, figures, and existing data.

\* Focus on suggestions that amplify the manuscript's strengths and maximize the impact of its contributions. Instead of flagging claims as overstated, suggest how to rephrase or restructure the narrative to make them more compelling based on the existing evidence.\*\*

# Inputs
## Research Idea:
```markdown
{RESEARCH IDEA}
```

Note that idea_text represents a preliminary hypothesis and may not necessarily align with the experiments that were eventually performed.

## Experiment Summaries (JSON format)
Refer to the experiment summaries contained in baseline_summary.json, improved_research_summary.json, hyperparam_ablation_summary.json and component_ablation_summary.json.
Important: The experiment results in hyperparam_ablation_summary.json and component_ablation_summary.json are summarized in the tables hyperparam_ablation_summary_table.tex and component_ablation_summary_table.tex.
In these .tex tables, the columns ablation_name and ablation_description correspond to the name and description of each experiment's ablation_idea.
Do not directly copy these descriptions into the .tex tables. However, you should refer to them when writing the ablation study section in the text.
You may modify how the non-numerical content is written in the .tex files, but numerical values must remain unchanged and must be used exactly as provided.
(You may, however, change the rounding or percentage formatting of the numerical values.)

## Available Plots (list of filenames)
```
{PLOT LIST}
```

## Plot Descriptions (from VLM)
```
{PLOT DESCRIPTION}
```

To better understand the idea, experiments, and results, please also refer to:
- baseline_tex/ and baseline_code/baseline.py as the baseline TeX and code, respectively. When writing up the baseline method, you must follow its method section.
- baseline_tex/baseline_bibtex.txt contains the baseline paper's bibliographic information and title - read this file to identify the baseline method being compared against.
- proposed_method_code/improved_proposed_method.py as the proposed method's code implementation (note that the proposed method should be referenced in improved_proposed_method.py rather than proposed_method.py)
- method_section.tex (pre-written Method section with detailed technical analysis)

# Output Generation
Based on your review, generate the following two files.

---
## Part 1: Feedback Report (Markdown)
- **File to Generate:** `reflection_1_feedback.md`
- **Format:** Markdown
- **Content:** A comprehensive report detailing the findings from all four criteria of the `Core Task`.

### Markdown Report Template

### 1. Results of Falsification & Inconsistency Check
- [ ] No significant inconsistencies or signs of falsification were found.
- [ ] The following discrepancies or concerns were identified:
- (e.g., Table 1 reports an accuracy of 92.5%, but the JSON data shows an average of 91.8%. This suggests a potential cherry-picking of the best seed's result.)
- (e.g., The text describes the result in Figure 3 as a "significant improvement," but comparison with the VLM description indicates it is within the margin of error. The claim is overstated.)
- (e.g., The method section states a dropout rate of 0.5, but the implementation in the source code uses 0.3.)

### 2. Results of Guideline Adherence Check
- [ ] The manuscript generally adheres to the provided guidelines.
- [ ] The following deviations from the guidelines were found:
- (e.g., The font used for equations is inconsistent. Please follow the instructions in CLAUDE.md.)
- (e.g., The rule that figure captions must be placed below the figure has not been followed.)

### 3. Results of Missing Citations Check
- [ ] No significant missing citations were identified.
- [ ] The following sections may require additional citations:
- (e.g., The second paragraph of the Introduction discusses the basic Transformer architecture but lacks a citation to the original paper, "Attention Is All You Need.")
- (e.g., A citation should be added for the iNaturalist dataset used in the experiments, referencing the paper that introduced it.)

### 4. Section-by-Section Improvement Suggestions
- **Abstract:** (e.g., The background description is currently too long. The abstract would be more impactful if it started with a concise one-sentence summary of the contribution and included key quantitative results.)
- **Introduction:** (e.g., The problem statement is clear, but the motivation could be strengthened by better explaining why this problem is important and what the broader impact of solving it would be.)
- **Related Work:** (e.g., Rather than simply listing prior studies, structure the section thematically. Clearly explaining how your work differs from or builds upon key papers will better highlight your unique contribution.)
- **Method:** (e.g., The pseudocode in Algorithm 1 is dense. Consider simplifying it to show only the main steps and moving detailed explanations to the main text to improve readability.)
- **Experimental Setup:** (e.g., For reproducibility, it would be beneficial to include a table summarizing all key hyperparameters. Justifying the choice of evaluation metrics and baselines would also strengthen the paper's credibility.)
- **Experiments:** (e.g., Instead of just listing results, add more analysis immediately after each result to discuss *why* it occurred. This would add significant depth to the paper.)
- **Ablation Study:** (e.g., It's unclear which components of your method contribute most to the performance gain. An ablation study that systematically analyzes the impact of each component would provide valuable insights and strengthen your claims.)
- **Conclusion:** (e.g., Providing more specific details on the study's limitations and future directions

would demonstrate scientific rigor and foresight.)

---
## Part 2: Required Citations List (JSON)
- **File to Generate:** 'reflection_1_required_bib.json'
- **Format:** JSON (A list of strings)
- **Content:** A machine-readable list of the descriptive strings for each missing citation identified in 'Core Task 3'. Each string in the list should be a clear instruction for a search tool.

### JSON Format Example
```json
[
"The original Transformer paper, 'Attention Is All You Need' by Vaswani et al.",
"The paper that introduced the iNaturalist dataset."
]
```

---

Prompt for Feedback Revision

# Role
You are an expert writing assistant who accurately incorporates feedback to improve a manuscript. Your task is to faithfully execute the requested revisions to enhance the overall quality of the text.

# Instructions
Revise the ./template.tex strictly according to the contents of the Feedback Report found at reflection_1_feedback.md. Implement all the changes and suggestions noted in the feedback and output the complete, revised manuscript. Do not introduce new information or interpretations that are not specified in the feedback. When the feedback points out missing citations, you must add appropriate \citep{} commands in the main text. Refer to the Newly Acquired Citations (BibTeX format).

**Requests Beyond the Scope of Existing Data (CRITICAL):**
* If a feedback requests a new experiment, a comparison with data you do not have, or an analysis that is impossible with the provided information, **DO NOT INVENT DATA OR RESULTS.**
* Instead, address the comment *in the manuscript's text* by:
* **First, attempting to reframe the point as a strength or a deliberate choice using the principles in Strategy #5. As a last resort, if this is not feasible, proceed by:**
* Acknowledging the value and validity of the feedback's suggestion.
* Politely stating that it falls outside the scope of the current study's experiments.

# Inputs
## Research Idea:
```markdown
{RESEARCH IDEA}
```
Note that idea_text represents a preliminary hypothesis and may not necessarily align with the experiments that were eventually performed.

## Experiment Summaries (JSON format)
Refer to the experiment summaries contained in baseline_summary.json, improved_research_summary.json, hyperparam_ablation_summary.json and component_ablation_summary.json.
Important: The experiment results in hyperparam_ablation_summary.json and compo-

nent_ablation_summary.json are summarized in the tables hyperparam_ablation_summary_table.tex and component_ablation_summary_table.tex.

In these .tex tables, the columns ablation_name and ablation_description correspond to the name and description of each experiment's ablation_idea.

Do not directly copy these descriptions into the .tex tables. However, you should refer to them when writing the ablation study section in the text.

You may modify how the non-numerical content is written in the .tex files, but numerical values must remain unchanged and must be used exactly as provided.

(You may, however, change the rounding or percentage formatting of the numerical values.)

## Available Plots (list of filenames)
```
{PLOT LIST}
```

## Plot Descriptions (from VLM)
```
{PLOT DESCRIPTION}
```

## Newly Acquired Citations (BibTeX format)

To better understand the idea, experiments, and results, please also refer to:
- baseline_tex/ and baseline_code/baseline.py as the baseline TeX and code, respectively. When writing up the baseline method, you must follow its method section.
- baseline_tex/baseline_bibtex.txt contains the baseline paper's bibliographic information and title - read this file to identify the baseline method being compared against.
- proposed_method_code/improved_proposed_method.py as the proposed method's code implementation (note that the proposed method should be referenced in improved_proposed_method.py rather than proposed_method.py)
- method_section.tex (pre-written Method section with detailed technical analysis)

Please note: For the bibliography, do not use the \begin{filecontents}{references.bib} environment. Instead, all citations should refer to an external file named references.bib.

---

### CLAUDE.md for the writing agent

You are an ambitious AI researcher who is looking to publish a paper that will contribute significantly to the field.

Ensure that the paper is scientifically accurate, objective, and truthful. Accurately report the experimental results, even if they are negative or inconclusive.

You are planning to submit to a top-tier ML conference, which has guidelines:
- The main paper is limited to 8 pages, including all figures and tables, but excluding references, the impact statement, and optional appendices. In general, try to use the available space and include all relevant information.
- The main paper should be single-column format, while the appendices can be in single-column format. When in single column format, make sure that tables and figures are correctly placed.
- Do not change the overall style which is mandated by the conference. Keep to the current method of including the references.bib file.
- Do not remove the
graphicspath directive or no figures will be found.
- Do not add 'Acknowledgements' section to the paper.
- Use a single backslash (\) for LaTeX commands instead of a double backslash (\\).

Here are some tips for each section of the paper:

- **Title**:
- Title should be catchy and informative. It should give a good idea of what the paper is about.
- Try to keep it under 2 lines.

- **Abstract**:
- TL;DR of the paper.
- What are we trying to do and why is it relevant?
- Make sure the abstract reads smoothly and is well-motivated. This should be one continuous paragraph.

- **Introduction**:
- Longer version of the Abstract, i.e., an overview of the entire paper.
- Provide context to the study and explain its relevance.
- Highlight how the approach effectively addresses the research question or problem. Also, provide an intuitive explanation of why this method works well.
- Summarize your contributions, highlighting pertinent findings, insights, or proposed methods.

- **Related Work**:
- Academic siblings of our work, i.e., alternative attempts in literature at trying to address the same or similar problems.
- Compare and contrast their approach with yours, noting key differences or similarities.
- Ensure proper citations are provided.

- **Method**:
- Clearly detail what you propose to do and why. If your study aims to address certain hypotheses, describe them and how your method is constructed to test them.

- **Experimental Setup**:
- Explain how you tested your method or hypothesis.
- Describe necessary details such as data, environment, and baselines, but omit hardware details unless explicitly mentioned.

- **Experiments**:
- Present the results truthfully according to the data you have. If outcomes are not as expected, discuss it transparently.
- Include comparisons to baselines if available, and only include analyses supported by genuine data.
- Try to include all relevant plots and tables. Consider combining multiple plots into one figure if they are related.
- In tables, please bold the best result among the compared methods. There is no need to bold the proposed method if it is not the best.

- **Conclusion**:
- Summarize the entire paper, including key strengths or findings.
- Results are strong, highlight how they might address the research problem.

- **Appendix**:
- Place for supplementary material that did not fit in the main paper.
- Add more information and details (hyperparameters, algorithms, etc.) in the supplementary material.
- Add more plots and tables in the supplementary material. Make sure that this information is not already covered in the main paper.

- When checking for duplicate figures, be sure to also review their descriptions to catch cases where different figures convey the same information. For example, one figure might present aggregated training accuracy as a single line plot with a shaded standard deviation (e.g., aggregated_training_accuracy.png), while another (per_seed_training_accuracy.png) shows the same data as three separate line plots.

Ensure you are always writing good compilable LaTeX code. Common mistakes that should be fixed include:
- LaTeX syntax errors (unenclosed math, unmatched braces, etc.).
- Duplicate figure labels or references.
- Unescaped special characters: & % $ # _ { } ~^\
- Proper table/figure closure.
- Do not hallucinate new citations or any results not in the logs.

Ensure proper citation usage:
- Always include references within \begin{filecontents}{references.bib} ... \end{filecontents}, even if they haven't changed from the previous round.
- **Before citing any paper, ALWAYS read the references.bib file first to find the correct citation key. Do NOT create new fictional bibtex entries.**
- Do not make any changes to reference.bib
- **Verify all citation keys match exactly with those in references.bib before using them in the text.**
- Note that the baseline BibTeX entry is located near the bottom of references.bib
- Each section (especially Related Work) should have multiple citations.

## Notes
- Do not include formatted mathematical formulas or equations in the Abstract. However, including key numerical results (e.g., "an accuracy of 92.5%") and mentioning mathematical concepts by name is perfectly acceptable.
- During the reflection process, the abstract should not be significantly altered.
- Do not include results in the Method section.
- Do not include Limitation sections and Future Work sections.
- Use '%‘ to display a literal percent sign, as a standard '%‘ will be treated as a comment.

# C   Detail of Agents4Science Reviews

This section presents detailed reviews from Agents4Science. In Section 5, we focused on reviews that address the fundamental limitations of the Jr. AI Scientist. In contrast, this section provides detailed reviews to clarify what is required for AI Scientist–generated papers to be accepted at future Agents4Science conferences. (If the goal is just acceptance, including human intervention is the fastest way.) As noted in Section 5, the papers submitted were generated by an earlier version of the Jr. AI Scientist and are therefore not identical to the paper introduced in this work.

## C.1   LoCoOp Extention Paper

**Reviewer 1**

**Review Score**

**Score:** 3: Borderline reject

**Reviewer Comments**

The paper proposes Nuisance-Prompt Tuning (NPT) for few-shot OOD detection with CLIP, introducing a learnable nuisance prompt, attention-weighted patch supervision, and a margin loss to separate nuisance and class prompts. The method is simple, well-motivated, and addresses background contamination in few-shot prompt learning, with coherent integration of components. Empirical gains are consistent on three of four OOD datasets, with the largest improvement on iNaturalist, but overall improvements are modest (AUROC +0.006; FPR95 -0.028 absolute). The evaluation is limited in breadth (only one backbone, few baselines, no multi-shot results in main text), and lacks statistical rigor (no error bars, single-seed reporting). The reliance on CLS attention as a proxy for foreground/background is plausible but unvalidated. Some ablation results raise questions about training confounds. The inference scheme is sensible but under-explored. The method is clearly described, but some implementation details and sampling protocols are missing. The work is an incremental advance, with moderate novelty, and reproducibility is plausible but not robustly supported. Ethics and limitations are briefly discussed. Actionable suggestions include expanding evaluation (more backbones, baselines, multi-shot, error bars), validating the attention assumption, analyzing SUN degradation, clarifying implementation, and exploring alternative inference schemes. Overall, the paper is a clear and reasonable contribution with promising intuitions, but the empirical evidence and breadth fall short of top-tier standards. Recommend rejection in current form, with a path to acceptance after expanded, statistically robust evaluation and stronger baselines/analyses.

## Reviewer 2

**Review Score**

**Score:** 5: Accept

**Reviewer Comments**

Strengths:
Clear motivation and problem formulation, with a precise critique of prior work. Technically sound, novel, and elegant method, with clever use of CLIP's self-attention and a well-designed loss function. Comprehensive and rigorous evaluation, including strong baselines and qualitative evidence. Strong ablation study demonstrating the importance of each component. High clarity and readability throughout the paper.
Weaknesses:
Lack of statistical significance testing due to single-run experiments; error bars or confidence intervals are not reported. Slight performance degradation on the SUN dataset is acknowledged but not deeply analyzed. Limited discussion of limitations; a dedicated section would be beneficial. Overall, this is a high-quality paper with a solid technical contribution, rigorous validation, and excellent clarity. The weaknesses are minor and do not detract significantly from the overall strength of the work. The paper makes a clear and valuable contribution to the field and is well-suited for publication at a top-tier conference. Strongly recommended for acceptance.

## Reviewer 3

**Review Score**

**Score:** 3: Borderline reject

**Reviewer Comments**

This paper introduces Nuisance-Prompt Tuning (NPT), a method for few-shot out-of-distribution (OOD) detection that addresses background contamination in prompt learning approaches. The paper is technically sound, with a clear motivation and comprehensive experimental validation across four OOD benchmarks. The method uses a dedicated 'nuisance' prompt, attention-weighted supervision, and margin-based repulsion, and is well-presented with thorough ablation studies. However, the improvements over baselines are modest, with only small gains in FPR95 and AUROC, and there is performance degradation on the SUN dataset, raising concerns about robustness and generalizability. The approach is incremental, introducing additional hyperparameters despite claims of reduced tuning complexity, and the novelty is mainly in the combination of existing techniques. The paper is well-written and reproducible, but the AI-generated nature of the work raises questions about the validity of experimental claims. Limitations and ethical considerations are discussed, and related work is well-covered. Specific concerns include small improvement margins, unexplained performance drops, questionable claims about threshold brittleness, limited analysis of computational overhead, and dataset-dependent effectiveness. Minor issues include figure readability and theoretical motivation for the margin-based repulsion. Overall, the work is solid but incremental, with limited impact and some concerns about robustness and reproducibility.

## C.2 Min-K%++ Extention Paper

**Reviewer 1**

**Review Score**

**Score:** 2: Reject

**Reviewer Comments**

The paper introduces Multi-Layer Concentration Analysis (MLCA) to improve pre-training data detection by augmenting Min-K%++ with distribution-shape features computed at multiple layers. While the motivation is clear and the method is described transparently, the review identifies several significant weaknesses: (1) lack of calibration/validation for per-layer distributions, (2) errors and ambiguities in aggregation and normalization formulas, (3) inconsistent application of the method across architectures (multi-layer only for Mamba, not Pythia), (4) modest and mixed empirical gains with insufficient statistical rigor (no confidence intervals, inconsistent reporting), (5) hand-chosen fusion weights without proper justification or cross-validation, (6) limited baselines (only Min-K%++), and (7) incomplete reporting of experimental details and reproducibility. The contribution is seen as incremental, with originality limited by the use of standard features and lack of novel multi-layer probing. The review recommends rejection, suggesting that a revised version addressing calibration, formula correctness, fairer comparisons, stronger baselines, and more rigorous evaluation could be a solid contribution.

**Reviewer 2**

**Review Score**

**Score:** 3: Borderline reject

**Reviewer Comments**

Strengths:
Novel insight that SSMs like Mamba benefit more from multi-layer analysis than Transformers, suggesting fundamental architectural differences. Well-motivated and technically sound approach,

building on strong baselines and using theoretically grounded features. Strong empirical results for Mamba, with clear visualizations and meaningful AUROC improvements. Clear writing, good structure, and a dedicated limitations section. Weaknesses:

Critically flawed evaluation for the Transformer (Pythia): the comparison is invalid due to a "simplified" analysis for Pythia, lacking transparency and rigor, undermining claims about architectural differences. Inconsistent hyperparameter selection: main results use a suboptimal value, weakening confidence in the findings. Limited robustness: small sample sizes, single runs, and no statistical significance tests make it hard to assess the reliability of the reported gains. Overall, the paper presents a promising idea and strong results for Mamba, but major experimental flaws—especially regarding the Transformer evaluation and hyperparameter inconsistency—outweigh the reasons to accept. The paper is not ready for publication in its current form, but could be reconsidered if these issues are addressed.

**Reviewer 3**

### Review Score

**Score:** 3: Borderline reject

### Reviewer Comments

This paper presents Multi-Layer Concentration Analysis for enhancing pre-training data detection in large language models. The work is technically sound, building on the Min-K%++ baseline and introducing concentration features (Shannon entropy, Gini coefficient, top-k concentration, effective vocabulary size) from multiple network layers. The mathematical formulations are correct and the experimental methodology is reasonable. However, improvements are modest (especially for Pythia, 0-1 percentage points AUROC), there is no statistical significance testing due to single runs, and the theoretical justification for multi-layer analysis could be stronger. The paper is generally well-written and organized, with clear methodology and effective figures, though some technical details (like layer selection and feature aggregation weights) could be clearer. The significance is limited, with incremental improvements and the most notable gains for Mamba (up to 1.9 percentage points). The architectural insights are interesting but not groundbreaking. The originality lies in combining multi-layer analysis with concentration features, but the core ideas are not particularly novel individually. The paper provides sufficient detail for reproduction, though the lack of error bars and single runs limit reproducibility assessment. Ethical considerations are adequately discussed, focusing on data privacy and copyright. The related work section is comprehensive and citations are appropriate. Major concerns include modest improvements, single runs without statistical testing, shallow theoretical understanding, and limited exploration of architectural differences. Minor issues include arbitrary hyperparameter choices, limited computational overhead analysis, and some unsupported claims about architectures. Overall, the paper addresses an important problem and shows consistent if modest improvements, representing an incremental advance rather than a significant breakthrough.

## C.3 GL-MCM Extention Paper

**Reviewer 1**

### Review Score

**Score:** 3: Borderline reject

**Reviewer Comments**

Strengths:
The method is simple, clear, and training-free, with minimal overhead. Uses a principled uncertainty signal to address max-pooling weaknesses. Qualitative evidence is provided via score distributions.
Weaknesses:
Evaluation is on very small subsets (100–500 images), lacking error bars and statistical testing, which limits reliability and invites sampling variance. Baselines are limited to GL-MCM; no comparisons to other zero-shot OOD methods (e.g., CLIPN, ZOC, MCM). The method is sensitive to the $\alpha$ parameter, with no automatic selection mechanism, raising deployment risk. Inconsistency in claims: Table 1 claims "consistent improvements," but SUN AUROC decreases while FPR95 improves. Subset selection details (randomization, seeds) are unspecified, limiting interpretability and reproducibility. The "first" claim of an information-theoretic framework is overstated, as related ideas exist. Recommendations: Expand evaluation to full ImageNet-OOD benchmarks, use multiple random seeds, and report confidence intervals. Compare against more zero-shot OOD methods under a standardized protocol. Provide sensitivity analyses and investigate automatic $\alpha$ selection. Clarify subset selection and compute details; include runtime benchmarks. Add qualitative visualizations of entropy weights on images. Correct or qualify the "consistent improvements" claim and discuss the SUN AUROC trade-off. Consider alternative uncertainty signals and report ablations. If claiming theoretical grounding, provide supporting analysis. Overall, the idea is neat and potentially useful, but the current empirical evidence is too limited for acceptance at a high-standard venue. Strengthening evaluation and comparisons would improve the paper's credibility and impact.

**Reviewer 2**

**Review Score**

**Score:** 3: Borderline reject

**Reviewer Comments**

This paper introduces Entropy-Weighted Local Concept Matching (ELCM), a novel, theoretically-motivated method for zero-shot out-of-distribution (OOD) detection using vision-language models. The main contribution is an entropy-based weighting scheme for aggregating patch-level predictions, addressing the limitations of max-pooling in prior work (GL-MCM). The paper is well-written, clearly motivated, and demonstrates consistent improvements over the GL-MCM baseline on several OOD datasets, with notable reductions in FPR95. The authors are transparent about the method's limitations.

However, the experimental validation is insufficient: the evaluation is limited to a single baseline (GL-MCM), with no comparison to other state-of-the-art methods, making it difficult to assess the true significance of the approach. The experiments are conducted on small samples without statistical robustness (no error bars or multiple runs), and the method is highly sensitive to a key hyperparameter. Additionally, the impact of several engineering enhancements is not disentangled from the core contribution due to a lack of detailed ablation studies.

Overall, while the idea is promising and the presentation is strong, the paper's empirical evidence is too weak to support acceptance. I recommend rejection in its current form, but encourage the authors to address the experimental shortcomings and resubmit, as the core idea could underpin a strong future paper.

**Reviewer 3**

**Review Score**

**Score:** 3: Borderline reject

**Reviewer Comments**

This paper presents Entropy-Weighted Local Concept Matching (ELCM), an improvement to GL-MCM for zero-shot out-of-distribution detection in vision-language models. The technical approach is sound and theoretically motivated, using Shannon entropy to weight patch-level contributions, but the improvement is modest (AUROC from 0.9129 to 0.9188, FPR95 from 0.3495 to 0.2975) and introduces hyperparameter sensitivity. The implementation includes ad-hoc components that undermine the theoretical elegance. The paper is well-written and clearly structured, though some implementation details are relegated to the appendix. The impact is limited, as improvements are modest and only compared to GL-MCM, not other methods. The originality lies in applying entropy-based weighting, but the work is incremental. Reproducibility is reasonable, with code provided, but evaluation is limited to 100 images per dataset. The authors are transparent about limitations, including hyperparameter sensitivity and limited baseline comparisons. Related work is adequate but could be broader. Major concerns include limited evaluation scope, modest improvements, hyperparameter sensitivity, and ad-hoc enhancements. Strengths are theoretical motivation, honest evaluation, consistent improvements, and clear presentation. Overall, the paper is a solid incremental contribution but does not make a significant impact due to modest improvements and practical limitations.

