# OpenReview forum: "Jr. AI Scientist and Its Risk Report: Autonomous Scientific Exploration from a Baseline Paper"
_TMLR — Accepted by TMLR_

### Review · Reviewer_MZWK · 2025-11-29

**Summary Of Contributions:**

The paper presents Jr. AI Scientist, which is an autonomous agent system designed to mimic the research workflow of a student researcher. The authors also include a risk report and evaluations, including AI reviewers, author inspections, and Agents4Science reviews.

**Additional Comments:**

The paper presents a promising and methodologically rich system. The risk reporting perspective is especially valuable.

**Audience:**

Yes

**Audience Explanation:**

It might be good to include how TMLR audience can benefit from this Jr. AI scientist work. In the paper, the contribution focuses on Jr. AI scientist development and risk evaluations. It might be good to build a toolbox or website for this type of Jr. AI scientist would potentially increase participation and enhance the work’s practical impact.

**Broader Impact Concerns:**

This work could become an impactful reference for AI scientists' system design. It's important to explain how generalizable this type of AI scientist system is to the whole ML or general science community.

**Claims And Evidence:**

Yes

**Claims Explanation:**

Yes, but some claims need a more clearly defined scope, evaluation rigor, and workflow justification.

**Requested Changes:**

1. Clarity of Framing:

The framing often changes between presenting Jr. AI Scientist as a “novice student analog,” a “state-of-the-art autonomous agent,” and an “upper bound demonstration” of current AI scientist capabilities. These three framings may imply different success criteria. What is the clear definition of Jr. AI scientist? The narrative would benefit from explicitly defining the intended scope early. Otherwise, the evaluation becomes difficult to interpret.

In the introduction section, it will be great to reorganize to explicitly define what “capability,” “upper bound,” and “novice workflow emulation” mean in measurable terms. This will also clarify the evaluation criteria.

2. Clarity of Framework and Pipeline:
Figure 2 contains two-stage 3 framework. It's a bit confusing for readers to differentiate between these two stage 3 modules, though they are all ablation stages. It might be beneficial to combine them into one big stage 3, and draw out two boxes to show one is about hyperparameter ablation and the other is component ablation.

For iterations, why does Stage 1 have 12 iterations and Stage 2 have 50 iterations? There is no justification for these numbers. How do authors get to this number of 12 but not bigger?

Additionally, the experiments rely heavily on a single coding agent (Claude Code) and a single LLM family (o4-mini, GPT-4o reviewers). Cross-model validation is necessary for broader claims. If it's not feasible to run with other agents, perhaps adding a limitation section and stating this limitation would also be sufficient.

3. Evaluations:
It's interesting to see the author's evaluation relies heavily on a mix of: AI-based review scoring (DeepReviewer), Author-led curation of “best papers,” and Agents4Science reviews. In my opinion, the use of “best-of-several generated papers” may introduce significant selection bias.

3.1. It will be great to provide the distribution of review scores across all generated papers. For example, create a table to obtain the full distributions of scores (e.g., min / median / max).

3.2. The DeepReviewer system can be unstable and sensitive to formatting, writing style, and non-semantic features. It will be clear to state the DeepReviewer reliability, variance, and known biases.

3.3. For the Agents4Science evaluation, please also provide excerpts, reviewer comments, or score summaries in the appendix or manuscript so that the evaluation criteria are clear and reproducible for future researchers.

4. Some minor comments: The novelty verification via Semantic Scholar remains fragile. In the paper Limitation section, it will be great to mention the limitations of LLM-based literature checking.

---

> ### Author Response · Authors · 2026-01-05
> **Author Rebuttal 1**
>
> We sincerely appreciate that the reviewer has taken the time to review our work thoroughly.
>
> We have carefully considered your suggestions and updated the manuscript accordingly, with changes highlighted in magenta. Below, we provide detailed responses to each point.
>
> ### **W1: Clarity of Framing. The relationship between the definition of the Jr. AI Scientist and the notions of a “novice student analog,” a “state-of-the-art autonomous agent,” and an “upper-bound demonstration.”**
>
> Thanks for the question. A Jr. AI Scientist is defined as an AI Scientist that is given baseline resources and focuses on extending the baseline. This setting is novice student analog, where a student builds upon an existing paper which the mentor gave. By evaluating the Jr. AI Scientist, we demonstrate that it represents a state-of-the-art AI Scientist. Therefore, the expressions “a state-of-the-art autonomous agent” and an “upper-bound demonstration” are not the definition of a Jr. AI Scientist itself, but rather observations drawn from our evaluation results. We have clarified this definition more explicitly in the Introduction.
>
> ### **W2: Two stage 3 modules in Figure2 is confusing.  It might be beneficial to combine them into one big stage 3**
>
> Thank you for the comment. As pointed out, we have revised the figure for Stage 3 accordingly in the revision.
>
> ### **W3: How the number of iterations in the experimental stage is determined**
>
> The number of iterations was chosen empirically, but with a clear rationale. Stage 1 is intended solely for implementing the proposed ideas and does not aim at performance improvement, unlike Stage 2 (Iterative Improvement); therefore, fewer iterations are sufficient for Stage 1. In contrast, Stage 2 aims to improve performance beyond the baseline and requires a larger number of iterations. Therefore, we set the number of iterations to 50.
>
> While increasing the number of iterations in Stage 2 beyond 50 could potentially lead to higher-performing methods, it would substantially increase computational costs. Therefore, we place it outside the scope of this work.
>
> ### **W4: The experiments rely heavily on a single coding agent and a single LLM family, and lack cross-model validation.**
>
> Thanks for the question. As pointed out, we built our AI Scientist on top of a single coding agent (Claude Code) and a single LLM family. We did not explore combining other LLMs and coding agents to further improve performance. We leave this direction for future work. In the revision, we have added a limitation section and included this discussion.
>
> ### **W5.1: Selection criteria for author paper selection are unclear**
>
> Thanks for the question. As the selection criterion, an author who is highly familiar with the baseline paper carefully examined the content of the six generated papers for each baseline, the obtained results, and the code, and selected the paper that were considered to be of high quality.
>
> ### **W5.2: Scores of all generated papers**
>
> Thanks for the valuable suggestion. We have included the scores of all 18 (6 papers * 3 baselines) generated papers in the revision. Overall, the average score across all 18 papers is 5.30. While this is lower than the 5.75 reported in Table 2, it is still higher than those of existing AI-generated papers. Moreover, although the authors’ selections well align with DeepReviewer’s scores, the highest-scoring papers are not always selected. Upon detailed human inspection, we found that the highest-scoring papers contain more hallucinations, highlighting the importance of author inspection and honest reporting of hallucinations in addition to review scores. We have included further details in Sec. 4.2.
>
> ### **W6: Potential Bias of DeepReviewer**
>
> Thank you for the valuable feedback. As noted in the DeepReviewer paper, DeepReviewer shows stronger alignment with human judgments than existing AI reviewers. However, like any AI (and human) reviewer, it may still be influenced by bias, which is discussed in the Ethical Considerations section in the DeepReviewer paper. This is not a limitation unique to DeepReviewer but a broader challenge for the research field in AI reviewers. We believe that reducing bias and building more reliable reviewers is an important challenge for the research community.

---

> > ### Author Response · Authors · 2026-01-05
> > **Author Rebuttal 2**
> >
> > ### **W7:  It would be better to include detailed evaluation results at Agents4Science**.
> >
> > Thank you for the valuable feedback. To support the community’s future efforts toward Agents4Science, we have included our detailed review results.
> >
> > ###  **W8: Some minor comments: The novelty verification via Semantic Scholar remains fragile. In the Limitation section, it will be great to mention the limitations of LLM-based literature checking.**
> >
> > Thank you for the comment. Indeed, developing novelty-checking methods beyond Semantic Scholar is an important challenge for the research community. We have explicitly discussed this point in the limitations section.
> >
> > ###  **Audience Interest: how TMLR audience can benefit from this Jr. AI scientist work.**
> >
> > Thanks for the question. As Reviewer **mujM** also pointed out, we believe that this paper is of interest to a broad range of researchers, including (1) researchers in agentic AI and automated discovery, (2) researchers in ethical AI and safety, and (3) researchers specializing in peer review.
> >
> > ### **Audience Interest: Building a toolbox or website for this type of Jr. AI scientist would potentially increase participation and enhance the work’s practical impact.**
> >
> > Thank you for the suggestion. We plan to gradually make a toolbox. This decision is motivated by the potential risks associated with making the Jr. AI Scientist easy to use.
> > We agree that building a toolbox makes the Jr. AI Scientist accessible to a wider range of users, which could increase its practical impact. However, as discussed in this paper, AI Scientist systems, including the Jr. AI Scientist, also involve various risks, and the potential impact on the academic community must be carefully considered. Therefore, we plan to build such a toolbox after carefully evaluating the risks that this system may pose to the research community.  We have added this discussion in Appendix A.

---

> > > ### Comment · Reviewer_MZWK · 2026-02-09
> > > **Thanks for the clarifications.**
> > >
> > > Thanks for the rebuttal context. This information is very helpful.
> > >
> > > The clarification on W5.2: Scores of all generated papers are interesting. It would be great to illustrate it in the discussion section.
> > >
> > > And W4: The experiments rely heavily on a single coding agent and a single LLM family, and lack cross-model validation, which is also helpful. I don't think author(s) solve this question, but including this in future work could help researchers to explore different coding agents' capabilities.
> > >
> > > Generally speaking, great rebuttal!

---

### Review · Reviewer_mujM · 2025-12-18

**Summary Of Contributions:**

Key Contributions:
- Development of a complex multi-file AI Scientist system that significantly outperforms existing benchmarks from other recent works (5.75 vs 4.50)
- Generation of papers using 3 high-quality publications by making the AI agent follow realistic research workflow steps.
- A thorough risk report on various areas where human intervention and expertise is still required to ensure the integrity of the scientific research process.

Weaknesses:
- The paper reports scores based on 3 paper generation experiments. The authors claim that they have generated several papers but only evaluating the highest quality ones without providing any details as to what constitutes as "several" on Page 9.
- The system is dependent on human provided baseline paper and code artifacts for idea generation albeit this is of least concern.

**Additional Comments:**

N/A

**Audience:**

Yes

**Audience Explanation:**

The paper targets a problem in the emerging field of agentic AI led research which readers of the below groups would be interested in:

1. Researchers in Agentic AI and Automated Discovery
The primary audience would be those tracking the evolution of "AI Scientists." The paper introduces a system that mimics human research workflows, handles multi-file codebases and achieves state-of-the-art review scores. In automated assessments, it earned an average rating of 5.75, substantially outperforming systems like AI Scientist-v1 (3.30) and AI Scientist-v2 (2.75).

2. Ethical AI and Safety Researchers
The paper highlights review-score hacking and integrity issues with citation accuracy, hallucination of facts, and the fabrication of descriptions in autonomous research workflows.

3. Peer Review and Meta-Learning Specialists
Researchers focused on improving AI reviewers or understanding the peer-review process would find the paper's evaluation methodology valuable. It uses DeepReviewer, author-led evaluations, and actual submissions to the Agents4Science conference (where AI acts as both author and reviewer) to validate its findings.

**Broader Impact Concerns:**

There are no Broader Impact Concerns with this paper as the authors are very forthright in reporting AI use and on the clarity of the experimentation conducted. The paper also presents a risk report outlining various different risks of using AI to conduct scientific research which would be very helpful in studying the impact of such work.

**Claims And Evidence:**

Yes

**Claims Explanation:**

Despite of the weaknesses, the paper still advances AI agent driven AI research benchmarks significantly and also incorporates a feedback loop for LLM based review step towards the end of the research workflow. It also outlines various risks associated with their system where human intervention and expertise is required.

**Requested Changes:**

- I would like the authors to provide more information around how many papers they generated in order to come up with the three papers provided as supplemental material and if possible also quantify the cost of doing so. This will provide the paper a new axis on which future work can be conducted in spirit of AI efficiency.

---

> ### Author Response · Authors · 2026-01-05
> **Author Rebuttal**
>
> We sincerely appreciate that the reviewer has taken the time to review our work thoroughly.
>
> We have carefully considered your suggestions and updated the manuscript accordingly, with changes highlighted in magenta. Below, we provide detailed responses to each point.
>
> ### **W1.1 and RC1.1: More detailed information on the generated papers**
>
> Thanks for the valuable question. For each baseline, we generated six candidate papers. An author who is highly familiar with the corresponding baseline papers carefully examined the content of the six generated papers, their experimental results, and the associated code, and selected those considered to be of high quality. We have added these details in Section 4.1.
>
> ### **W1.2 and RC1.2: Cost for generating papers**
>
> Regarding cost, Claude Code is the most resource-intensive agent in our pipeline. However, by using the Claude Code Max plan (USD 200 per month), it is possible to run two experiments in parallel within the usage limit, enabling paper generation without incurring substantial cost.  We have added these details in Section 4.1.
>
> ### **W2 (Minor): The system is dependent on human provided baseline paper and code artifacts for idea generation albeit this is of least concern.**
>
> Thank you for the comment. Although full automation of the entire research workflow is a promising direction, key challenges such as paper selection and baseline reproduction remain highly difficult, even for humans. Therefore, this work focuses on evaluating the capabilities and risks of an AI Scientist under minimal human-provided resources

---

### Review · Reviewer_bvD1 · 2025-12-23

**Summary Of Contributions:**

This manuscript investigates the potential of AI scientists for the task of expanding a baseline paper. First, by analyzing the baseline both in terms of manuscript and code. Then identifying a gap. Latter generating ideas to cross that gap. And finally, by analyzing by reflection the proposed idea and improving them.

Strengths
- The paper is well written, transmitting clear ideas, following logical sequences, and technically sound.
- The overall methodology seems appropriate as it follows the human approach to conduct research. Nonetheless, the details of each phase can be better explained.
- The results show that the proposed method outperforms previous methods.
- Authors acknowledge the limitations of the current state of the model.

Weaknesses
- The proposed method is requieres a lot of human interventions, for instance to manually define the abstract, input the code, or define the captions of figures, which makes it difficult to identify whether the method is simply a sequence of calls to LLM's or a real chain of thoughts in itself.
- There are several figures that are not cited in the main text. They simply appear.
- There is a passage that reads "If all nodes fail, the system selects the next nodes...", which is contradictory in itself. If all nodes have failed, then there is no next node to be selected.
- The computation details of the scores used for comparison are not provided.

**Audience:**

Yes

**Audience Explanation:**

The topic is of interest to the TMLR community, and the proposed methodology seems coherent and appropriate. Moreover, the results show that the proposed model outperforms previous ones.

**Broader Impact Concerns:**

There is no concern about ethical implications, as authors have covered all requirements.

**Claims And Evidence:**

No

**Claims Explanation:**

Must claims about the design and experimentation sound logical and needed. However, there is no evidence about the results, other than Table 2.

**Requested Changes:**

- Section 3.3.2 requires a clearer explanation.
- Section 3.4.1 requires a clearer explanation.
- Prompts used for each stage must be provided for reproducibility.
- All acronyms must be defined at their first appearance, e.g., OOD detection.
- The computation details of the scores used for comparison must be provided.

---

> ### Author Response · Authors · 2026-01-05
> **Author Rebuttal**
>
> We sincerely appreciate that the reviewer has taken the time to review our work thoroughly.
>
> We have carefully considered your suggestions and updated the manuscript accordingly, with changes highlighted in magenta.
> Below, we provide detailed responses to each point.
>
> ### **W1: The proposed method requires substantial human intervention (e.g., defining the abstract, providing code, and specifying figure captions), making it unclear whether it constitutes a genuine reasoning process or merely a sequence of LLM calls.**
> We would like to clarify several assumptions. As shown in Fig. 1, the Jr. AI Scientist requires only the baseline code, paper, and LaTeX as inputs, and generates the abstract and figure captions without any human intervention. Providing only the baseline code, paper, and LaTeX as inputs minimizes human involvement. With the minimum human involvement, Jr. AI Scientist formulates its own ideas, tests them through experimentation, and writes the full paper. We therefore consider this to constitute a genuine reasoning process. We have added further details on the human intervention in Section 3.4.1.
>
> ### **W2: There are several figures that are not cited in the main text. They simply appear.**
>
> Thank you for pointing it out. In the revised version, we have correctly cited Fig. 1 and Fig. 4 in the main text.
>
> ### **W3: There is a passage that reads "If all nodes fail, the system selects the next nodes...", which is contradictory in itself. If all nodes have failed, then there is no next node to be selected.**
>
> We apologize for the unclear description. The total number of nodes that exist during the experimental phase is different from the number of nodes managed in each iteration (four). To clarify this point, we added the following description in the revised version:　"During execution, four experimental nodes are selected. If all currently running nodes fail, the system selects the next nodes, either by initializing new nodes from the baseline code or by debugging previously generated buggy codebases."
>
>
> ### **W4, RC5: The computation details of the scores used for comparison are not provided.**
>
> Thank you for the valuable comment. DeepReviewer is a 14B-parameter LLM trained on the DeepReview-13K dataset to perform structured, human-like paper assessment.
> It evaluates technical soundness, experimental validity, and logical consistency using standardized scores.  DeepReviewer shows stronger alignment with human judgments than existing AI reviewers, enabling efficient and reliable evaluation.  The experiments for this evaluation were conducted on a single A100 80GB GPU.
>
> We have added further details on DeepReviewer and the computational setup used in our experiments to Section 4.2.
>
>
> ### **RC1, RC2: Section 3.3.2 requires a clearer explanation and Section 3.4.1 requires a clearer explanation.**
>
> Thanks for pointing it out. In Section 3.3.2, we have added a clearer description of the node selection mechanism. In Section 3.4.1, we have added a clearer description specifying which inputs need to be prepared by humans.
>
> ### **RC3: Prompts used for each stage must be provided for reproducibility.**
>
> Thank you for the suggestion. We plan to gradually make them available. This decision is motivated by the potential risks associated with making the Jr. AI Scientist readily reproducible. As discussed in this paper, AI Scientist systems raise various concerns. The primary goal of this work is to present the current capabilities and risks of AI Scientists in order to foster a deeper understanding within the research community. We therefore believe that it is necessary to carefully assess these risks before proceeding with a broader release. We have added this discussion in Appendix A.
>
> ### **RC4: All acronyms must be defined at their first appearance, e.g., OOD detection.**
>
> Thank you for the suggestion. We added the definition of the OOD detection task at its first occurrence in the Introduction.

---

### Author Response · Authors · 2026-01-05
**General Response by Authors**

Dear Reviewers,

We thank all the reviewers for their valuable effort, feedback, and comments. We are happy to hear that reviewers recognized our paper's strengths: The topic is of interest to the TMLR community (**bvD1**, **mujM, MZWK**), the methodology is coherent, appropriate, and promising (**bvD1**, **MZWK**), reporting various risks is beneficial (**mujM, MZWK**), and the paper advances AI agent-driven AI research (**mujM**).

In response to the reviewers’ feedback, we have revised the manuscript. All modifications are highlighted in magenta. We have written the responses to the concerns of each reviewer in each thread.

We greatly appreciate the time, thoughtful consideration, and considerable effort to reviewing our paper.

Sincerely,

The Authors

---

### Decision · Action_Editor_X29h · 2026-02-15

**Recommendation:** Accept with minor revision

**Additional Comments:**

Appendix A mentions a gradual release of code and prompts, based on possibly risk (yet to be evaluated), this goes against open science and reproducibility of this paper, considering the possible low risks, I suggest that the appendix contain at least the most important prompts used in the Jr AI Scientist and the CLAUDE.md file, to make sure future research can reproduce the results and study the weaknesses in the proposed method. Note that keeping details hidden (security through obscurity) has historically not led to the prevention of such risks.

I also suggest to mention in the abstract that the improvement in review score is based on scores given by DeepReviewer, as the reader can be misled to think these are human review scores, which they are not.

Finally, I suggest that the final version of this paper should contain a Broader Impact Statement / Ethics Statement, where the authors discuss the ethical issues of doing research with AI agents and the possible risks in a broader setting.

**Audience:**

Yes

**Audience Explanation:**

There is a clear audience for this paper given the broad interest of using LLMs for scientific discovery, this paper contributes with a workflow that instead of just generating papers with LLMs, it uses a workflow that generates implementation and then papers from a baseline paper, which is validated and experimented by a human, so while the results are not massively impressive, the overall workflow is novel and likely to be of interest to TMLR's audience.

All reviewers agree that they are satisfied with the audience interest.

**Claims And Evidence:**

Yes

**Claims Explanation:**

This paper proposed a workflow for developing a Jr AI Scientist based on a baseline paper and several steps that reflect the human scientific workflow to generate new ideas and new scientific publications, with a decent evaluation using LLM reviews viea DeepReviewer in the context of a scientific workshop.

The authors claim that their Jr AI Scientist works better than the baselines, and it overcomes some limitations of previous AI scientist systems, and this is clearly supported by the evidence. A possible criticism of the evidence is that review scores for evaluation are generated by an LLM (Deep Reviewer), which can introduce a degree of bias, human reviewers could have been better but the review process itself is noisy and imperfect.

All reviewers agree that they are satisfied with the claims.

---

> ### Author Response · Authors · 2026-02-21
> **Camera-ready Submission**
>
> Dear Editor,
>
> We sincerely appreciate the decision of “Accept with Minor Revision.”
>
> In the camera-ready version, we have made the following revisions in accordance with your requests:
>
> - Added the Main Prompts and CLAUDE.md, which are now included in Appendix B.
>
> - Clarified in the Abstract that the improvement in review score is based on scores assigned by DeepReviewer.
>
> - Added a Broader Impact Statement, in which we discuss the ethical considerations of conducting research with AI agents and the potential risks in a broader context.
>
> Thanks very much for your consideration.
>
> Sincerely, The Authors